# The influence of ice sheets on temperature during the past 38 million years inferred from a one-dimensional ice sheet-climate model

Lennert B. Stap[1,2], Roderik S. W. van de Wal[1], Bas de Boer[1], Richard Bintanja[3], and Lucas J. Lourens[4]

[1]Institute for Marine and Atmospheric research Utrecht (IMAU), Utrecht University, Princetonplein 5, 3584 CC Utrecht, The Netherlands
[2]Alfred Wegener Institute, Helmholtz Centre for Polar and Marine Research (AWI), Bussestrasse 24, 27570 Bremerhaven, Germany
[3]Royal Netherlands Meteorological Institute (KNMI), Utrechtseweg 297, 3731 GA, De Bilt, The Netherlands
[4]Department of Earth Sciences, Faculty of Geosciences, Utrecht University, Heidelberglaan 2, 3584 CS Utrecht, The Netherlands

*Correspondence to:* Lennert B. Stap (L.B.Stap@uu.nl)

**Abstract.** Since the inception of the Antarctic ice sheet at the Eocene-Oligocene Transition ($\sim$34 Myr ago), land ice has played a crucial role in Earth's climate. Through feedbacks in the climate system, land ice variability modifies atmospheric temperature changes induced by orbital, topographical and greenhouse gas variations. Quantification of these feedbacks on long time scales has hitherto scarcely been undertaken. In this study, we use a zonally averaged energy balance climate model bi-directionally

coupled to a one-dimensional ice sheet model, capturing the ice-albedo and surface-height-temperature feedbacks. Potentially important transient changes in topographic boundary conditions by tectonics and erosion are not taken into account, but briefly discussed. The relative simplicity of the coupled model allows us to perform integrations over the past 38 Myr in a fully transient fashion, using a benthic oxygen isotope record as forcing to inversely simulate $CO_2$. Firstly, we find that the results of the simulations over the past 5 Myr are dependent on whether the model run is started at 5 or 38 Myr ago. This is because

the relation between $CO_2$ and temperature is subject to hysteresis. When the climate cools from very high $CO_2$ levels, as in the longer transient 38 Myr run, temperatures in the lower $CO_2$ range of the past 5 Myr are higher than when the climate is initialized at low temperatures. Consequently, the modeled $CO_2$ concentrations depend on the initial state. Taking the realistic warm initialization into account we come to a best estimate of $CO_2$, temperature, ice volume-equivalent sea level and benthic $\delta^{18}O$ over the past 38 Myr. Secondly, we study the influence of ice sheets on the evolution of global temperature and polar

amplification by comparing runs with ice sheet-climate interaction switched on and off. By passing only albedo or surface height changes to the climate model, we can distinguish the separate effects of the ice-albedo and surface-height-temperature feedbacks. We find that ice volume variability has a strong enhancing effect on atmospheric temperature changes, particularly in the regions where the ice sheets are located. As a result, polar amplification in the Northern Hemisphere decreases towards warmer climates as there is little land ice left to melt. Conversely, decay of the Antarctic ice sheet increases polar amplification

in the Southern Hemisphere in the high-$CO_2$ regime. Our results also show that in cooler climates than the pre-industrial, the ice-albedo feedback predominates the surface-height-temperature feedback, while in warmer climates they are more equal in strength.

# 1 Introduction

The most abundant information source with the highest resolution on Cenozoic global climate change are stacked benthic oxygen isotope ($\delta^{18}$O) records (Lisiecki and Raymo, 2005; Zachos et al., 2008; Cramer et al., 2009), which have been well-studied and statistically analysed (e.g. Mudelsee et al., 2014). The benthic $\delta^{18}$O signal is known to be comprised of two factors (e.g. Chappell and Shackleton, 1986): 1) the deep-sea temperature, and 2) the volume of land ice on Earth. An additional independent record of either one is therefore required to separate the signal into its individual constituents. Deep-sea temperature records can be reconstructed based on the $\mathrm{Mg/Ca}$ proxy (Lear et al., 2000; Sosdian and Rosenthal, 2009; Elderfield et al., 2012), but a global average deep-sea temperature is hard to obtain. Sea level records are also available, but are subject to the same problem of inferring a global mean (Miller et al., 2005; Kominz et al., 2008; Rohling et al., 2014). Studies using sea level records face the additional challenge of converting local sea level to ice volume, which is not straightforward mainly because of dynamic topography (Mitrovica and Milne, 2003; Kendall et al., 2005). Alternatively, calculation of benthic $\delta^{18}$O can be incorporated in coupled ice sheet-climate models, by using parameterisations of the contribution of deep-sea temperature (Duplessy et al., 2002), and the isotopic content of ice sheets (Cuffey, 2000). Hitherto, studies using this approach have mostly focused on relatively short time intervals surrounding important climatic events, such as the Eocene-Oligocene Transition (33.9 Myr ago; Tigchelaar et al. (2011); Ladant et al. (2014); Wilson et al. (2013)), the Mid-Miocene Climatic Optimum (13.9 Myr ago; Langebroek et al. (2010); Gasson et al. (2016)), and the Pliocene-Pleistocene Transition (2.6 Myr ago, Willeit et al. (2015)), using models of varying complexity. These studies have provided valuable information because they have simulated these key events in great detail. However, they do not to describe climate change on multi-million year time scales, and prove consistency by transiently simulating multiple events using the same set-up. This is mainly due to insufficient computer power. An effort was made to simulate the past 3 Myr using a model of reduced complexity (Berger et al., 1999), but they did not include the Southern Hemisphere and the Antarctic ice sheet in their model. Here, we will use a coupling between an energy balance global climate model and a one-dimensional ice sheet model of all major ice sheets, to perform transient simulations over the past 38 Myr. By focusing on the long term evolution of climate, we provide a complementary approach to snap-shot and short time slice experiments of more complex models.

Our model approach builds on the inverse routine to derive atmospheric temperature from benthic $\delta^{18}$O, that was introduced by Oerlemans (2004). This methodology was consequently developed further to force stand-alone three-dimensional ice sheet models over the past 1 Myr (Bintanja et al., 2005; De Boer et al., 2013), the past 3 Myr (Bintanja and Van de Wal, 2008), and the past 5 Myr (De Boer et al., 2014). With this inverse routine, the past 40 Myr were simulated (De Boer et al., 2010) and further analysed (De Boer et al., 2012), using a one-dimensional ice sheet model that calculates all land ice on Earth. In Stap et al. (2014), this ice sheet model was coupled to a zonally averaged energy balance climate model (Bintanja, 1997), and run over the past 800 kyr forced by a compiled ice core $CO_2$ record (Petit et al., 1999; Siegenthaler et al., 2005; Lüthi et al., 2008). Inclusion of a climate model added $CO_2$ to global temperature and sea level as an integrated component of the simulated system. In addition, it rendered the possibility of investigating ice sheet-climate interactions, specifically the ice-albedo and surface-height-temperature feedbacks. Furthermore, instead of annual mean and globally uniform temperature perturbations

to present-day climate, seasonal meridional temperature distributions were used to force the different ice sheets. In a subsequent study, the inverse routine was transformed to yield $CO_2$ concentrations using the benthic $\delta^{18}O$ as input, making $CO_2$ a prognostic variable (Stap et al., 2016a). The resulting values were used to force the coupled model over the past 5 Myr. In Stap et al. (2016b), the model was run over the period 38 to 10 Myr ago, and the influence of Antarctic topographic changes on the simulated $CO_2$ was investigated. In this study, however, changes in topographic boundary conditions are not included, although their effect is briefly discussed.

Here, we will use this coupled ice sheet-climate model forced by benthic $\delta^{18}O$ to transiently simulate the entire past 38 Myr. We recognise that our simulation of $CO_2$ may be improved in subsequent studies that include geological processes that are still missing in our model setup. For instance tectonics leading to mountain uplift (Kutzbach et al., 1993) and closure of sea ways (Kennett, 1977; Toggweiler and Bjornsson, 2000; Hamon et al., 2013), erosion (Wilson et al., 2012; Gasson et al., 2015; Stap et al., 2016b), and vegetation changes (Knorr et al., 2011; Liakka et al., 2014; Hamon et al., 2012) may have affected the climate system during the past 38 Myr. Here, we will purely focus on the influence of ice sheets on the climate, in particular the relation between $CO_2$ and temperature, during this time. Earlier studies using more complex stand-alone ice sheet models and coupled ice sheet-climate models have for example determined the $CO_2$ thresholds for glaciation of Antarctica (DeConto and Pollard, 2003; Langebroek et al., 2010; Ladant et al., 2014; Gasson et al., 2014) and the Northern Hemisphere (DeConto et al., 2008; Hamon et al., 2013). Furthermore, they have investigated the hysteresis in the relation between ice volume and $CO_2$ (Pollard and DeConto, 2005), as well as the behaviour of the Antarctic ice sheet during the Oligocene (Pollard et al., 2013) and Plio-Pleistocene (Pollard and DeConto, 2009). Our model represents reduced complexity in the model hierarchy (see e.g. Pollard (2010) for a discussion), and adds a long-term transient perspective on the evolution of ice sheets and the climate. It reconciles knowledge on benthic $\delta^{18}O$, $CO_2$, sea level and temperature. However, it lacks ice-shelf dynamics and sophisticated grounding line parameterisations that have been shown to be important at least on short time scales (Pollard and DeConto, 2012; Pollard et al., 2015). Therefore, this study should be seen as a first step in the direction of simulating multi-million year time spans with coupled ice sheet-climate models, aiming to combine temperature, sea level and $CO_2$ in one framework. As such, its goal is two-fold. Firstly, as a precursory study, we attempt to identify interesting phenomena and potential obstacles, and set a reference simulation over the past 38 Myr. When results of more sophisticated models are achieved, they can be compared to ours to see which features appear in the full hierarchy of models and which are specific to more comprehensive models including more physics. Secondly, we perform multiple 38-Myr integrations of our coupled model with ice-sheet climate interactions switched on and off. This allows us to quantify the effect of these interactions on global temperature perturbations and polar amplification, distinguishing between the ice-albedo and the surface-height-temperature feedback.

## 2  Model

We use the same simplified coupled ice sheet-climate model setup as Stap et al. (2016b) (Fig. 1). The climate component is a zonally averaged energy balance climate model (based on North, 1975; Bintanja, 1997) with $5°$ latitudinal and 0.5 day temporal

resolution. It includes a simple ocean model that has 6 vertical layers and mimics meridional ocean circulation with varying strength based on the density difference between the polar and equatorial waters. Sea ice cover is calculated thermodynamically at $1.25°$ resolution. The climate model provides the local monthly temperature input ($T$) to the mass balance module of one-dimensional models of the five major Cenozoic ice sheets (Eurasian, North American, Greenland, West Antarctic and East Antarctic ice sheets) (De Boer et al., 2010). Herein, accumulation follows as a temperature-dependent fraction of precipitation ($P$):

$$P = P_0 e^{0.04T - R/R_c}, \tag{1}$$

where $P_0$ is present-day precipitation, $R$ is ice-sheet radius, and $R_c$ is an ice-sheet dependent critical radius. An insolation-temperature melt formulation is used to calculate ablation:

$$M = [10T + 0.513(1 - \alpha)Q + C_{abl}]/100. \tag{2}$$

Here, $\alpha$ is surface albedo, and $Q$ local radiation obtained from Laskar et al. (2004). Ice-sheet dependent tuning factors $C_{abl}$ determine the thresholds for which ablation starts. The tuning targets and consequent values for $C_{abl}$ have been described in Stap et al. (2014). The influence of this parameter for inception of the East Antarctic ice sheet has been discussed in Stap et al. (2016b), who showed that changing the value from -30 to -5 results in glacial inception at roughly 450 ppm instead of 700 ppm. The ice sheet models calculate the surface height change and ice sheet extent using the Shallow Ice Approximation with 15 to 25 km resolution depending on the ice-sheet (De Boer et al., 2010). This information is used to update the land ice fraction and surface height profile in the climate model for the next time step. Exchange of variables takes place every 500 model years, constituting the output-timestep of the coupled model. The isotopic content of the ice sheets is calculated using the parameterisation of Cuffey (2000):

$$\delta^{18}O_i = \delta^{18}O_{PD} + \beta_T \Delta T + \beta_Z \Delta Z, \tag{3}$$

where $\beta_T$ and $\beta_Z$ are ice-sheet dependent parameters (values listed in De Boer et al., 2010), that determine the influence of annual mean temperature ($\Delta T$) and surface height ($\Delta Z$) perturbatons with respect to present day. Present-day isotopic contents ($\delta^{18}O_{PD}$) match the modeled values of an earlier study by Lhomme et al. (2005). The modeled benthic $\delta^{18}O$ values follow from:

$$\delta^{18}O = [\delta^{18}O_b]_{PD} - \frac{\overline{\delta^{18}O_i V_i}}{V_o} + \left[ \frac{\overline{\delta^{18}O_i V_i}}{V_o} \right]_{PD} + \gamma \Delta T_o, \tag{4}$$

where $[\delta^{18}O_b]_{PD}$ is the observed present-day value of benthic $\delta^{18}O$, and $V_o$ and $V_i$ are the ocean and land ice volume. The final term on the right hand side quantifies the influence of deep-sea temperature change with respect to present day ($\Delta T_o$). Gain factor $\gamma$ is set to 0.28 ‰$K^{-1}$, taken from a paleotemperature equation (Duplessy et al., 2002). The coupled model is forced by insolation data (Laskar et al., 2004) and an inverse routine, which yields $CO_2$ concentrations from the difference between the modeled benthic $\delta^{18}O$ value and an observed value an output-timestep later (Stap et al., 2016a):

$$CO_2 = \overline{CO_2} * \exp[c * \{\delta^{18}O(t) - \delta^{18}O_{obs}(t + 0.5kyr)\}], \tag{5}$$

where $\overline{CO_2}$ is the mean $CO_2$ concentration of the preceding 15 kyr, and $c$ a strength-determining parameter (Stap et al., 2016a). For the observed benthic $\delta^{18}O$ values we use stacked records (Lisiecki and Raymo, 2005; Zachos et al., 2008), which effectively serve as model input. The radiative forcing anomaly with respect to present day is multiplied by a factor 1.3. to account for non-$CO_2$ greenhouse gases. This factor is based on analysis of $CH_4$ and $N_2O$ records over the past 800 kyr (see Stap et al., 2014), and used to account for the lack of knowledge on non-$CO_2$ greenhouse gases prior to that period. An increase or decrease in the relative contribution of non-$CO_2$ would need an opposing change in $CO_2$. The result of the coupled model consists of mutually consistent records of benthic $\delta^{18}O$, atmospheric $CO_2$, temperature, and ice-volume equivalent sea level.

## 3 Results and Discussion I: Long-term transient simulations

### 3.1 Hysteresis

We perform two model runs, one over the past 38 Myr (this run is called '38 Myr'), and one over the past 5 Myr (this run is called '5 Myr'). The 38-Myr run uses the stacked benthic $\delta^{18}O$ record of Zachos et al. (2008) as forcing. As a spin-up the model was initialized with a 1500 ppm $CO_2$ concentration for 50 kyr, and thereafter run for 2 Myr between 40 and 38 Myr ago using the inverse routine. This run is an extension of the reference run used in Stap et al. (2016b), to include the past 10 Myr. The record of Lisiecki and Raymo (2005) was used to force the 5-Myr run, after initializing the model for 100 kyr with 430 ppm $CO_2$. This run served as a reference run before in Stap et al. (2016a). When we compare the final 5 Myr of our 38-Myr simulation to our 5-Myr simulation, we notice that the 38-Myr simulation shows much lower $CO_2$ concentrations (Fig. 2b, green and blue lines). These contradicting results cannot be explained by the use of different forcing records - Zachos et al. (2008) for 38 Myr as opposed to Lisiecki and Raymo (2005) for 5 Myr - as these show similar values during this time (0.02 ‰ average difference, not shown). To explore the difference, we additionally conduct four pairs of experiments with the model in forward mode. In forward mode, we do not use the inverse routine, but force the model by a-priori designed $CO_2$ scenarios. We initialize the model using: a 450 ppm $CO_2$ concentration; no land ice; glacio-isostatically relaxed present-day topography. We force the model by changing the $CO_2$ input in steps of 50 ppm every 50 kyr. In one set of experiments (named 'up'), the $CO_2$ is first raised from 450 ppm to 1200 ppm, then lowered to 150 ppm, and increased again to 600 ppm. In the other set (named 'down'), the $CO_2$ is initially dropped from 450 to 150 ppm, then raised to 1200 ppm, and ultimately decreased again to 300 ppm. Insolation is kept at PD level throughout all these equilibrium experiments.

Starting at 450 ppm $CO_2$, the 'up' and 'down' runs show the same initial global temperature (Fig. 3a). However, in the 'down' run, where the $CO_2$ progresses stepwise downward first and then upward, the global temperatures at low ($< 450$ ppm) $CO_2$ values are approximately a degree lower than those in the 'up' run, where $CO_2$ is first raised and then lowered. When the 'down' run is integrated over another $CO_2$ cycle, it shows the same global temperatures as the 'up' run (not shown). This means that once the coupled model has experienced high $CO_2$ values during its run, as is the case in the 38-Myr run but not in the 5-Myr run, the climates at lower $CO_2$ are warmer. This has important consequences for the simulated $CO_2$ concentrations as they have to decline further to obtain similar temperatures, which is what happens in the transient 38-Myr simulation forced by

the inverse routine. The different branches in Fig. 3 are stable equilibria of the model. The model behaviour does not depend on the forcing rate: using 50 ppm/100 kyr, 50 ppm/50 kyr, or 100 ppm/100 kyr leads to the same results. This behaviour is a form of hysteresis as results depend on previous conditions of the model. The question now arises what the cause of this hysteresis is. The global temperature difference between the 'up' and 'down' run is 0.94 K at 150 ppm $CO_2$. The difference is generally larger at higher than at lower latitudes, peaking at 60° S and 80° N (Fig. 4, red line). When the ice sheet model is uncoupled, and the climate model is directly forced in the same manner but using PD ice sheets, the globally averaged temperature difference reduces to 0.69 K (Fig. 3b). Keeping the ocean overturning strength, a known source of hysteresis in other models (e.g. Zhang et al., 2014), fixed at PD also leads to a small reduction; the difference becomes 0.73 K (Fig. 3c). The combined effect of uncoupled ice and ocean overturning strength is still not sufficient to eliminate the hysteresis (Fig. 3d). Even when in addition sea ice and snow cover are kept constant, a small hysteresis (0.11 K difference at 150 ppm) is present. This difference is now more uniformly spread over the globe, albeit slightly larger in the Southern than in the Northern Hemisphere (Fig. 4, blue line). Having ruled out albedo and ocean overturning strength, which only act to enhance it, the source of the hysteresis is most likely the temperature dependency of atmospheric transparency to short and long wave radiation in the energy balance calculations (Bintanja, 1996). Unfortunately, disabling this dependency in the energy balance calculations leads to numerical instability, causing the model to crash shortly after initiation. However, its influence is suggested by the model converging to a smaller steady-state energy imbalance at the top of the atmosphere, when the model has previously experienced high $CO_2$ values. This is the case in both the fully coupled run (Suppl. Fig. 1a) and the run where surface albedo and ocean overturning strength are kept fixed (Suppl. Fig. 1b). An energy imbalance at the top of the atmosphere when the atmospheric and oceanic temperatures have equilibrated, means there is an energy leak. When this energy leak is positive, the outgoing radiation and therefore the radiative temperature of the Earth are underestimated. For steady-state PI conditions, most General Circulation Models (GCMs) show positive energy imbalances, typically below $2 \, \mathrm{Wm^{-2}}$ (Lucarini and Ragone, 2011). In our model, more energy is lost at higher $CO_2$ concentrations, meaning the model simulates increasingly too cold temperatures towards warmer climates. Therefore, the climate sensitivity is underestimated. However, the energy leak remains relatively small: 0 to 4.6 $\mathrm{Wm^{-2}}$ globally averaged for $CO_2$ values between 150 and 1200 ppm (Suppl. Fig. 1a). In the inverse mode runs, the energy imbalance is indeed also different: the energy imbalance is smaller in the 38-Myr run than in the 5-Myr run (Suppl. Fig. 2, blue and green lines), indicating that the modelled climate states in these runs are in different branches of the curve shown in Suppl. Fig. 1. Although the difference in energy imbalance is small between the different branches, it evidently has important implications for the simulated $CO_2$ in inverse mode.

## 3.2   Retuning: new reference simulation

Originally, the 5-Myr run was calibrated to an Antarctic ice core proxy-$CO_2$ record (Petit et al., 1999; Siegenthaler et al., 2005; Lüthi et al., 2008) over the past 800 kyr (Stap et al., 2016a). It shows negligible bias (-3.9 ppm) to that record. As a consequence of the hysteresis described in Sect. 3.1, the 38-Myr run shows much lower values than this proxy record with a -47.7 ppm bias (Fig. 2a, mind that here we show 1-kyr values instead of 40-kyr averages). The simulated $CO_2$ over the past

5 Myr in the 38-Myr run is also much lower than the hybrid proxy data-model reconstruction by Van de Wal et al. (2011) (Fig. 2b, black line). Therefore, we deduce that the $CO_2$ record of the 38-Myr run is not realistic over this period. To regain agreement with the ice core record, which we judge to be essential for a transient paleoclimate simulation, we define a new reference run in this study (new REF). In this new reference run we increase the cloud optical thickness parameter $\tau_{cl}$ from 3.11 to 3.41. We opt to alter this parameter because it was already used as a tuning parameter in the original climate model (Bintanja,

1997), and in the ice sheet-climate model coupling (Stap et al., 2014). Both values are physically plausible. Increasing $\tau_{cl}$ will lower the temperatures calculated by the climate model, such that for the same benthic $\delta^{18}O$ higher simulated $CO_2$ levels are obtained, in better agreement with the ice core record. However, this will also raise the threshold $CO_2$ level for the inception of the East Antarctic ice sheet (EAIS). By increasing the ablation threshold parameter $C_{abl}$ in the insolation-temperature-melt calculation (Eq. (2)) of the EAIS (from $-30$ to $-10$), this ice sheet glaciates at lower temperatures and therefore at lower $CO_2$

concentrations. This parameter was used in Stap et al. (2014) to match the $CO_2$ inception point for Antarctic glacial inception to the one found by DeConto and Pollard (2003) ($\sim 780$ ppm), and is also poorly constrained. Changing this parameter compensates the unintended $CO_2$ threshold increase. We force the model in the same way as the earlier 38-Myr run over the past 38 Myr using the stacked benthic record of Zachos et al. (2008) as forcing.

The $CO_2$ concentrations of the new reference simulation are shown in Fig. 2 by red lines. The simulated $CO_2$ levels right

before the Eocene-Oligocene Transition (EOT; $\sim 33.9$ Myr ago) and at the Middle-Miocene Climatic Optimum (MMCO; 17 to 15 Myr ago) are similarly high around 650 to 750 ppm, likewise as in the earlier 38-Myr simulation (Fig. 2c). In the time between these events, $CO_2$ in the new reference run is modestly higher, up to 100 ppm. This is because the deep-sea temperatures are lower at the same $CO_2$, and therefore contribute less to the $\delta^{18}O$ anomaly with respect to present day. Compensating for the lower deep-sea temperatures, higher $CO_2$ increases the $\delta^{18}O$ anomaly, by increasing both deep-sea

temperature and the contribution of ice volume, hence raising sea levels (not shown). After the MMCO, when the EAIS has stabilised to near-PD size, the new reference simulation shows higher $CO_2$ values than the earlier 38-Myr simulation. Over the past million years, the new reference simulation (Fig. 2a, red line) agrees much better with the 5-Myr run (Fig. 2a, green line) and with the ice core $CO_2$ record (Fig. 2a, cyan line); the bias with respect to this proxy-record is reduced to 13.6 ppm. Even after re-calibration, the simulated new reference $CO_2$ remains lower than in the 5-Myr run during the Pliocene and early

Pleistocene (Fig. 2b, green line), as a consequence of the hysteresis. Although it is more variable than the reconstruction based on a constant Earth System Sensitivity (ESS) by Van de Wal et al. (2011) (Fig. 2b, black line), the long-term means are now similar. It remains debatable whether the shorter 5-Myr run or the new reference simulation is the most veracious over the past 5 Myr. On the one hand, the long simulation carries a longer memory, which would be closer to the state of the actual climate system. Furthermore, the energy imbalance at the top of the atmosphere is not affected significantly by retuning the model, and remains lower in this simulation than in the 5-Myr run (Suppl. Fig. 2, red line). On the other hand, it is uncertain how accurately our climate model simulates very warm climates; the climate model is designed and tested for PD and LGM climates (Bintanja and Oerlemans, 1996). This argument favours the shorter 5-Myr run as the more trustworthy result, implying that $CO_2$ levels over the last 5 Myrs may have been up to 470 ppm.

## 3.3 Comparison to proxy CO$_2$ data

A comprehensive quantitative comparison between our CO$_2$ simulation and proxy data is hindered by scarcity and intermittency of data records. Nevertheless, in Fig. 5 we show the new reference CO$_2$ results together with proxy data over three periods where data is relatively abundant: the Late Pliocene (3.5 to 2.5 Myr ago), the Middle Miocene (18 to 13 Myr ago) and the Early Oligocene (35 to 30 Myr ago). The data is based on three important proxies (see also Beerling and Royer, 2011) : alkenones (Pagani et al., 1999, 2011; Badger et al., 2013), boron isotopes (Pearson et al., 2009; Foster et al., 2012; Greenop et al., 2014; Martínez-Botí et al., 2015) and stomata (Van der Burgh et al., 1993; Kürschner, 1996; Kürschner et al., 2008; Retallack, 2009). During the Late Pliocene, the simulated CO$_2$ is more variable than the alkenone data from Badger et al. (2013) (Fig. 5a). This discrepancy between our simulation and the alkenone proxy (Pagani et al., 1999) is persistent throughout the Miocene (not shown). The variability in our simulation is more in line with the boron isotope proxy (Martínez-Botí et al., 2015), although our simulated CO$_2$ values are lower between 3.2 and 2.8 Myr ago, closer to the alkenone estimates Badger et al. (2013). Modelling studies with GCMs generally use higher values for the Pliocene, for example 400 ppm in the forthcoming PlioMIP2 experiments (Haywood et al., 2016). Comparison to the boron isotope proxy over the Miocene is hampered by lack of data. It would be interesting to know if this proxy also shows larger variability during that epoch. During the Middle Miocene, our simulated CO$_2$ is considerably higher than all proxy data records (Fig. 5b). Contrarily, during the Early Oligocene, it is a little bit lower (Fig. 5c). As mentioned in Sect. 3.2, the CO$_2$ inception point for Antarctic glacial inception was tuned to the one found by DeConto and Pollard (2003) ($\sim$ 780 ppm). It is therefore in agreement with the range of Antarctic glaciation values found by Gasson et al. (2014), using combinations of an ice dynamical model coupled to seven climate models (Fig. 5c, yellow shading). Using a different tuning, our Middle Miocene values are closer to the observations, but then the Early Oligocene values are also much lower (see Stap et al., 2016b, for an analysis). In short, we are not able to simulate a difference in CO$_2$ between right before the Eocene-Oligocene Transition and during the Middle Miocene Climatic Optimum. As our CO$_2$ simulation is obtained using benthic $\delta^{18}O$, this could indicate a discrepancy between the $\delta^{18}O$ and CO$_2$ proxies. More likely, however, it is for the largest part due to processes missing in our model set-up. Over the course of millions of years, tectonics, erosion and vegetation have led to topography and albedo changes that affect the climate system. Contrary to the findings of Foster and Rohling (2013), these geological processes may have had a significant impact on the relation between CO$_2$, sea level and temperature (Gasson et al., 2016). Indeed, in Stap et al. (2016b) we showed that in our model erosion could lead to a changing relation between CO$_2$ and ice volume over time, bringing $\delta^{18}O$ and CO$_2$ in line with other proxy indicators arguing for the importance of erosion. Also, we use a uniform lapse rate correction of 6.5 K for height changes in our model, which is a potentially important simplification (Gasson et al., 2014; Botsyun et al., 2016). In the future, our simulation can be improved by including these processes. For now, with this limitation in mind, we will focus on the long-term interaction between ice sheets and the climate in our model.

## 4 Results and Discussion II: Ice sheet-climate interaction

Figure 6 shows the main results of the new reference run: benthic $\delta^{18}O$, atmospheric $CO_2$, ice-volume-equivalent sea level and global temperature. The latter two are largely similar to De Boer et al. (2010) and De Boer et al. (2012), where comparisons to proxy data and other modelling studies can also be found. For example, they showed satisfactory agreement with a sea level curve derived from Red Sea sediments (Rohling et al., 2009) and a Mg/Ca proxy data deep-sea temperature record (Sosdian and Rosenthal, 2009). In our model, the relation between temperature and ice volume can roughly be divided into three regimes (see also De Boer et al. (2010) and Van de Wal et al. (2011)): 1) at low $CO_2$ values, strong ice volume variability due to dynamic Northern Hemispheric ice sheets, 2) at intermediate $CO_2$ values, weaker variability, 3) at high $CO_2$ values, strong variability due to a dynamic Antarctic ice sheet. This constitutes a sigmoidal temperature-sea level relation. The data-analysis results of Gasson et al. (2012) show a similar shape, but with higher deep-sea temperature anomalies during the warmer-than-PI climates. This was also the case in an earlier study using the same ice sheet model, but with parameterised deep-sea temperatures (De Boer et al., 2010). The trend we model in sea surface temperatures is also lower then what is suggested by proxy data (Herbert et al., 2016) (not shown). The modeled relation between logarithmic $CO_2$ and sea level is also sigmoidal. This is in very good agreement with the results from Foster and Rohling (2013), who derived a functional relation between these quantities from a geological data perspective (Fig. 7). However, sea level during the stable middle regime is lower in our case. This is coherent with the modeling results of Gasson (2013) (Suppl. Fig. 3). Possibly, recent advances in ice sheet modeling (Pollard et al., 2015) can explain (part of) this difference between models and data. Furthermore, our highest $CO_2$ levels are slightly lower than the data shows. The modeled $CO_2$ threshold for Antarctic glaciation is highly dependent on the mass balance parametrisation (Stap et al., 2016b; Gasson, 2013), the climate model used (Gasson et al., 2014), and the Antarctic topography (Stap et al., 2016b; Gasson et al., 2014). In our case, this threshold is distinctly higher than for that for land ice in the Northern Hemisphere. This is mainly a consequence of the higher latitude of the Antarctic continent, and in line with earlier findings (DeConto et al., 2008).

Now, we will investigate the influence of ice sheet-climate interaction on polar amplification and Earth System Sensitivity (ESS). The ESS is defined as the global temperature response to a radiative forcing caused by changing $CO_2$, taking into account all climate feedbacks (PALAEOSENS Project Members, 2012). This radiative forcing by $CO_2$ is proportional to the logarithmic change of $CO_2$ (Myhre et al., 1998). In Fig. 8 (red dots), we therefore show the relation between global temperature anomalies from pre-industrial (PI) and the logarithm of $CO_2$ divided by a reference PI value of 280 ppm in our new reference run. Evidently, this relation is not constant, as in warm climates the global temperature increase for a given $CO_2$ increase is less strong. The slope of a least squares linear regression shows a value of 10.6 K for $\ln(CO_2/CO_{2,\text{ref}})$ values below 0 ($CO_2$ < 280 ppm: coldest climates), and 3.7 K for values above 0.69 ($CO_2$ > 560 ppm: warmest climates), a reduction of 65 %. These values are equivalent to 7.3 K and 2.6 K per $CO_2$ doubling respectively. Hence, ESS is not constant, in contrast with the implicit assumption in Van de Wal et al. (2011). In fact, in our model ESS is stronger at lower $CO_2$. This is similar to the findings of Hansen et al. (2013), who performed $CO_2$ doubling experiments using the simplified atmosphere-ocean model of Russell et al. (1995). However, their ESS decrease is less strong, as it drops from ∼6 per $CO_2$ doubling from 155 to 310 ppm

$CO_2$ to $\sim 5.5$ K from 620 to 1240 ppm (see their Fig. 8b). They eventually also find increased sensitivity again at very high $CO_2$ levels (2480 to 9920 ppm), which is outside the range we simulate during our time span.

In our ice uncoupled run, the slope of the relation between $CO_2$ and global temperature reduces by only 46 % from 5.6 K to 3.0 K going from the coldest to the warmest $\ln(CO_2/CO_{2,\mathrm{ref}})$ regime (Fig. 8, blue dots). In this case, the standard error of a linear regression through all data points is reduced by 58 % with respect to the fully coupled run, from 0.0050 K to 0.0021 K. The fact that the relation between $\ln(CO_2/CO_{2,\mathrm{ref}})$ and global temperature is better approximated by a linear fit when land ice is uncoupled means the $\log(CO_2)$-T relation is more linear. Hence, climate sensitivity is more constant. However, even when ice sheets are kept at PD level, the relation between logarithmic $CO_2$ and global temperature shows a declining slope (Fig. 8, blue dots). Therefore, decreased sensitivity at higher $CO_2$ is not only determined by reduced ice volume variability. This finding may be compared to the hybrid data-model results for climate sensitivity of Köhler et al. (2015), as well as to the modeled climate sensitivity of Friedrich et al. (2016). Köhler et al. (2015) investigated the relation between the radiative forcing of proxy-data $CO_2$ (Petit et al., 1999; Siegenthaler et al., 2005; Lüthi et al., 2008; Hönisch et al., 2009) - which is linearly related to logarithmic $CO_2$ (Myhre et al., 1998) - and modeled global temperature (De Boer et al., 2014, scaled). Friedrich et al. (2016) forced the intermediate complexity climate model LOVECLIM over the past 800 kyr using the ice core record (Petit et al., 1999; Siegenthaler et al., 2005; Lüthi et al., 2008) and a Northern Hemispheric ice sheet reconstruction (Ganopolski and Calov, 2011). The resulting climate sensitivity of these studies is opposite to ours, as they show increased climate sensitivity at higher $CO_2$ concentrations. These studies, however, consider a smaller range of $CO_2$. Furthermore, they calculate climate sensitivity in a different way. They do take into account ice volume variations, but compensate for their effect by adding their radiative forcing to the forcing induced by $CO_2$ variations (see PALAEOSENS Project Members, 2012). Implicitly assumed in their approach is that these radiative forcings have the same effect on temperature, which may not generally be the case (Yoshimori et al., 2011). The difference between the results of our model and these studies could point to contrasting strengths of the fast feedbacks in the climate system, which is material for future investigation. Our findings are in agreement with Ritz et al. (2011), who used a two-dimensional energy balance climate model that showed an increase of climate sensitivity from 3.0 K per $CO_2$ doubling at PI conditions to 4.3 K at LGM conditions.

The coldest global temperature anomaly in our results is amplified by 79 % (factor 1.79) if land ice changes are incorporated, by 50 % if only albedo is coupled (Fig. 8, black dots), and by 4 % if only surface height is coupled (Fig. 8, orange dots). The warmest anomaly is only increased by 21 % (factor 1.21) when ice is coupled, by 9 % when only albedo is coupled, and by 3 % when only surface height is coupled. This means the surface-height-temperature feedback becomes relatively more important in warmer climates.

The influence of ice sheets on the climate is strongest in the region where they are situated, leading to increased polar amplification. This is demonstrated by the relations between global temperature and Northern Hemispheric (40 to 80° N, Fig. 9a), and Antarctic temperature (60 to 90° S, Fig. 9b). In the Northern Hemisphere, the minimum local temperature with respect to PI is -2.0 K in the uncoupled case, and -9.5 K in the run with fully coupled land ice. When only the albedo or surface height changes are coupled, the Northern Hemispheric temperature anomaly reaches -6.4 K and -2.8 K low points respectively. Conversely, the amount of land ice lost in warmer climates is relatively small, as only the Greenland ice sheet ($\sim$ 7 m.s.l.e.) is

left to melt. Consequently, the Northern Hemispheric temperature is then not affected much by not including land ice changes. The remaining polar amplification in the Northern Hemisphere is hence mostly caused by other factors, such as sea ice and snow cover variability. In the Southern Hemisphere, the lowest temperature is similar for the coupled and uncoupled simulations, although it is achieved at a higher global temperature in the uncoupled case. These Southern Hemispheric temperatures are similarly low because the Antarctic ice sheet grows relatively little in size towards colder-than-PI conditions (see also Stap et al., 2014). When Antarctica is allowed to melt in warm climates, however, the local temperature increase becomes much stronger: 11.6 instead of 5.9 K with respect to PI. In these conditions, we find that coupling albedo changes leads to a maximum Antarctic temperature anomaly of only 7.0 K (Fig. 9b, black dots). When only surface height changes are coupled, this anomaly reaches 7.4 K (Fig. 9b, orange dots). This result implies that albedo changes are relatively less important in Antarctica than in the high latitudes of the Northern Hemisphere. The reason is that the Antarctic continent remains snow covered throughout most of the year when the land ice retreats, which reduces the albedo change (see also Stap et al., 2016a). Since temperature changes are strongest in the Southern Hemisphere in warmer-than-PI climates, this explains the increased relative importance of the surface-height-temperature feedback on ESS in these climates. The different response of the northern and southern high latitudes to $CO_2$ changes challenges the approach of De Boer et al. (2010) and De Boer et al. (2012), who reconstructed a single high-latitude temperature anomaly. Furthermore, their record cannot readily be translated to global conditions by a constant factor (as is done in e.g Martínez-Botí et al., 2015), because the conversion depends on the prevailing climate state. This problem with recalculating high latitude values in terms of global mean changes also holds for other local proxy data like marine, terrestrial or ice core records.

Finally, we compare the relation between global temperature and logarithmic $CO_2$ in three model runs with uncoupled ice (Fig. 10). In one run the ice sheets are kept at PD condition as before (now called PD ice, blue dots), in another one we use the LGM condition (LGM ice, black dots), and in the last one all ice is removed (no ice, red dots). Naturally, the more ice is present on Earth, the colder the climate becomes, so the LGM ice run is colder than the PD ice run, which in turn is colder than the no ice run. The difference between the PD ice and the no ice run is fairly uniform over the whole $CO_2$ range. The difference between the LGM ice and the no ice run, however, is larger in cold climates than in warm climates as it shrinks from ~2.8 to ~1.6 K. This is explained by the extra land ice in the LGM ice run cooling the climate and increasing the area on Earth covered by snow and sea ice. As a result of this area increase, the land surface has a higher albedo, which cools the climate further. In cold climates this effect is stronger because the snow- and sea ice-covered area grows more towards the equator, where there is more incoming solar radiation. Consequently, the albedo increase is more effective as it leads to absorption of more energy, and thus to a stronger temperature decrease.

## 5  Summary and conclusions

We have presented mutually consistent transient simulations of atmospheric $CO_2$ content, temperature, ice volume and benthic $\delta^{18}O$ over the past 38 million years. They were obtained using a coupling between a zonally averaged energy balance climate model and a one-dimensional ice sheet model. As forcing, we have used an inverse routine that yields atmospheric $CO_2$ from

an observed benthic $\delta^{18}$O record (Zachos et al., 2008). This allowed us to simulate periods before 800 kyr ago, for which ice core records are not available and $CO_2$ data are uncertain, scarce and intermittent (Beerling and Royer, 2011). Focusing on long-term interactions between land ice and climate, we have taken a complementary approach to snap-shot and short timescale

simulations that have been published before (e.g. Langebroek et al., 2010; Ladant et al., 2014; Gasson et al., 2014; Pollard et al., 2015). Our coupled model results represent an improvement upon the work of De Boer et al. (2010), who used the same ice-sheet model in stand-alone form to simulate the past 40 Myr (De Boer et al., 2010, 2012). The inclusion of a climate model has enabled us to simulate, and force the different ice sheets with, seasonal meridional temperature distributions instead of globally uniform perturbations to present-day climate with a fixed seasonal cycle. Nonetheless, we recognise that our coupled

ice sheet-model is relatively simple. It does not include ice-shelf dynamics and sophisticated grounding line parameterisations (Pollard and DeConto, 2012; Pollard et al., 2015). The physics in our model are not detailed enough to simulate sub-millennial climate dynamics, which could have important consequences on for instance tropical atmospheric circulation (Mulitza et al., 2008). However, more complex models, such as GCMs coupled to three-dimensional thermodynamic ice models, are as of yet not suitable to perform multi-million year integrations because of limited computer power. Our results therefore serve as

a reference, to which results of these more sophisticated models can be compared once they are achieved. This facilitates an analysis of which features appear in the full hierarchy of models and which are specific to more comprehensive models including more physics. Furthermore, by comparing our fully coupled simulation to model runs with the ice-albedo feedback, or the surface-height-temperature feedback, or both switched off, we have quantified the effect of ice-sheet climate interactions on Earth System Sensitivity (ESS) and polar amplification on long time scales.

In our model, the results for $CO_2$ concentrations lower than roughly 450 ppm depend on the transient evolution of $CO_2$. When during the run the model has previously experienced high $CO_2$ values, temperatures are higher than when this is not the case. This hysteresis is persistent even in runs without any change in albedo due to snow, sea-ice or permanent land-ice cover and without changes in ocean overturning strength. However, these factors do enhance it. The hysteresis is most likely a consequence of the temperature dependency of the energy balance calculations. This is suggested by the fact that the model

converges to a different steady-state energy imbalance at the top of the atmosphere, and therefore a different climate state, depending on the history of the run. It is still unknown whether this behaviour is also exhibited by other models. We therefore suggest that in the future, climate models should be tested for this behaviour by confronting them with high $CO_2$ values before simulating cooler climates. In our case, it leads to unrealistically low simulated $CO_2$ values over roughly the past 13 Myr in our long 38-Myr simulation when $CO_2$ levels are below 450 ppm, to compensate for the higher temperatures. We have

therefore retuned the model, by changing the uncertain cloud optical thickness parameter. This leads to lower temperatures at the same $CO_2$ levels, such that higher $CO_2$ concentrations are sufficient to obtain the same temperatures. After retuning, the simulated $CO_2$ over the past 800,000 years is in reasonable agreement with the ice-core record again. The relations between $CO_2$ and temperature, as well as between high- and low-latitude temperatures, are now different. However, the influence of ice sheet-climate interactions on ESS and polar amplification remains qualitatively the same.

     As was already demonstrated in Stap et al. (2016b), our model is unable to capture the difference in $CO_2$ suggested by proxy data between the time right before the Eocene-Oligocene Transition ($\sim$34 Myr ago) and during the Middle Miocene Climatic

Optimum ($\sim$15 Myr ago). This is because the forcing benthic $\delta^{18}$O values are similar during these times. Our simulation of $CO_2$ may be improved by extending the model with more aspects of the climate system, moving towards a full Earth System Model. Important aspects hitherto neglected in our model are the effects of dynamic vegetation (e.g. Knorr et al., 2011; Liakka et al., 2014; Hamon et al., 2012), and changing topographic boundary conditions as a result of tectonics and erosion (Wilson et al., 2012; Gasson et al., 2015; Stap et al., 2016b). Ultimately, the model could also be coupled to a carbon cycle model, e.g.

BICYCLE (Köhler and Fischer, 2004), in order to simulate climate using only insolation data as input.

     In our model, ice volume changes enhance the modeled effect of $CO_2$ on temperature via the ice-albedo and the surface-height-temperature feedbacks, particularly in the regions where the ice sheets are located. At low $CO_2$ values, the Northern Hemispheric ice sheets change in size, causing large fluctuations in the temperature on this hemisphere. The ice-albedo feedback is much stronger than the surface-height-temperature feedback in these conditions (see also Stap et al., 2014). This is

reflected in the Northern Hemispheric (40 to 80° N), Antarctic (60 to 90° S), and global temperature profiles. At intermediate $CO_2$ values, there is only weaker land ice volume variability, as in the Northern Hemisphere there is little land ice left to melt, and in the Southern Hemisphere it is not yet warm enough for deglaciation of Antarctica. Consequently, temperature changes are only minorly enhanced, both locally and globally. At high $CO_2$ values, the Antarctic ice sheet is more dynamic, so that temperature changes more strongly on the Southern Hemisphere. Here, the impact of the ice-albedo feedback is weaker, since most

of the continent remains snow-covered during large parts of the year when the ice sheet retreats. Hence, the surface-height-temperature feedback becomes relatively more important. When the ice sheets are kept constant, temperature perturbations are much less strong and more uniformly distributed over the globe.

*Author contributions.*   L.B.S., R.S.W.v.d.W. and B.d.B. designed the research. B.d.B., R.B., and L.B.S. developed the model. L.B.S. conducted the model runs and analysis, to which R.S.W.v.d.W. and B.d.B contributed. L.B.S. drafted the manuscript, with contributions from all

co-authors.

*Competing interests.*   The authors declare that they have no conflict of interest.

*Acknowledgements.*   We thank four anonymous reviewers for providing useful suggestions, which helped to improve the quality of the paper. We further thank Edward Gasson for sharing his data. Financial support for L.B. Stap was provided by the Netherlands Organisation of Scientific Research (NWO), grant NWO-ALW. Bas de Boer is funded by NWO Earth and Life Sciences (ALW), project 863.15.019. This

paper contributes to the gravity program "Reading the past to project the future", funded by the Netherlands Organisation for Scientific Research (NWO).

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

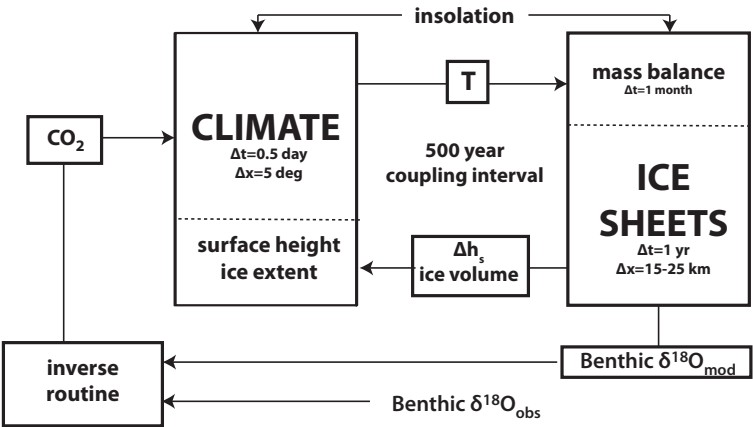

**Figure 1.** Schematic overview of the coupling of the zonally averaged energy balance climate model and the one-dimensional ice sheet model.

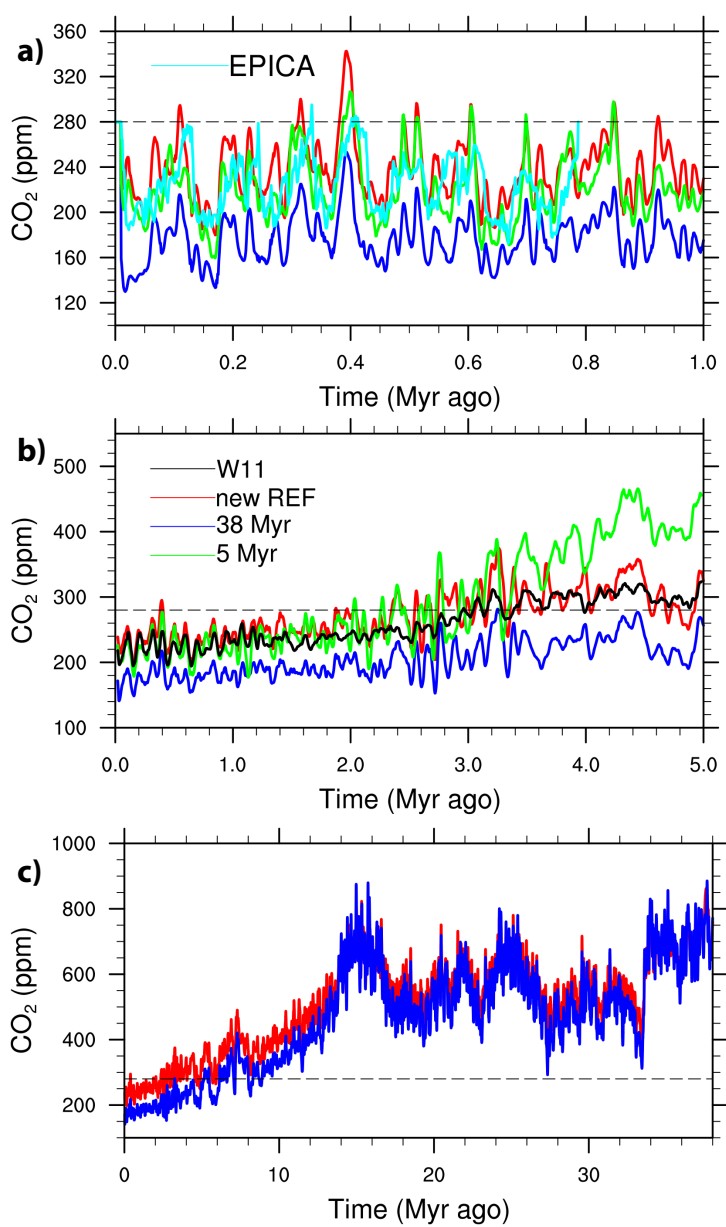

**Figure 2.** Atmospheric $CO_2$ over the past 1 Myr (a), the past 5 Myr (b) and the past 38 Myr (c). Shown are the 5-Myr run from Stap et al. (2016a) (green), the extended 38-Myr run from Stap et al. (2016b) (blue), the new reference run with altered cloud optical thickness (red), the hybrid proxy data-model reconstruction by Van de Wal et al. (2011) (W11; black), and the ice core record (Petit et al., 1999; Siegenthaler et al., 2005; Lüthi et al., 2008) (EPICA; cyan). The ice cores stem from Antarctica, the oldest values are from the EPICA Dome C core. Mind the differing y-scales. The dashed line shows the pre-industrial value (280 ppm). In panel (a) we do not show 40-kyr averages, but the 1-kyr output of the model.

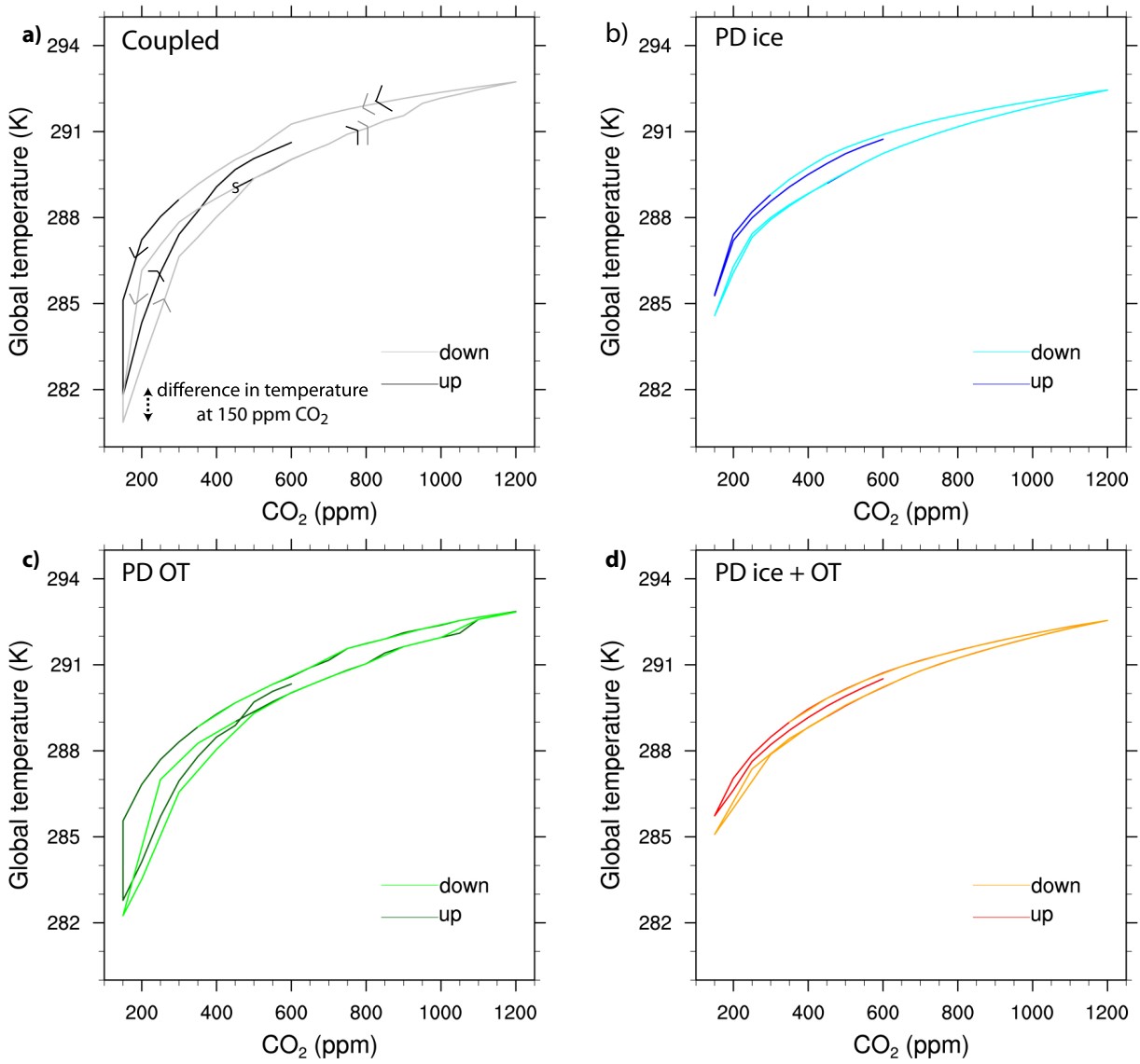

**Figure 3.** Relation between $CO_2$ and global temperature in the equilibrium runs. In (a), the fully coupled model output is shown. The starting point of the simulation at 450 ppm $CO_2$ is marked by an S, and the consequent evolution for both runs is marked by coloured arrows. The black line shows the up run, where $CO_2$ is increased first, the grey line shows the down run, where $CO_2$ is decreased first. At high $CO_2$ levels, the black line is overlaid by the grey line. In (b) the output with uncoupled ice (blue/cyan), in (c) with uncoupled ocean overturning strength (darkgreen/green) and in (d) with both these factors uncoupled (red/orange) are shown. The darker colors (blue, darkgreen, red) show the up runs, the lighter colors (cyan, green, orange) show the down runs. The startpoint and evolution are the same as in (a).

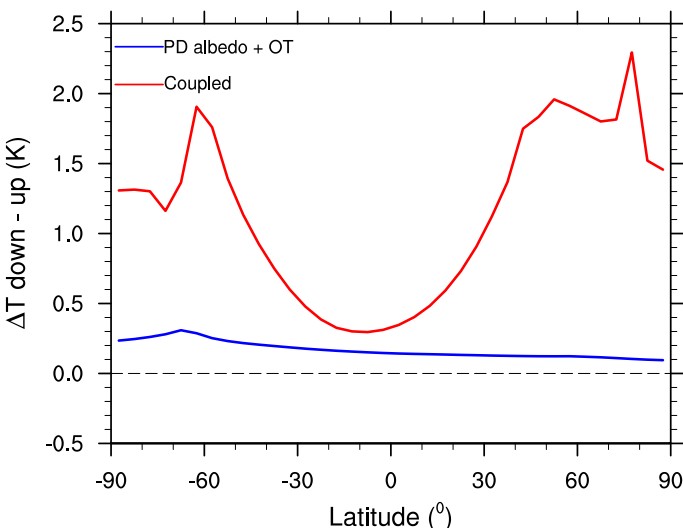

**Figure 4.** Atmospheric temperature difference at 150 ppm between the equilibrium runs (down - up), using the fully coupled set-up (red), and using fixed PD surface albedo and ocean overturning strength (blue).

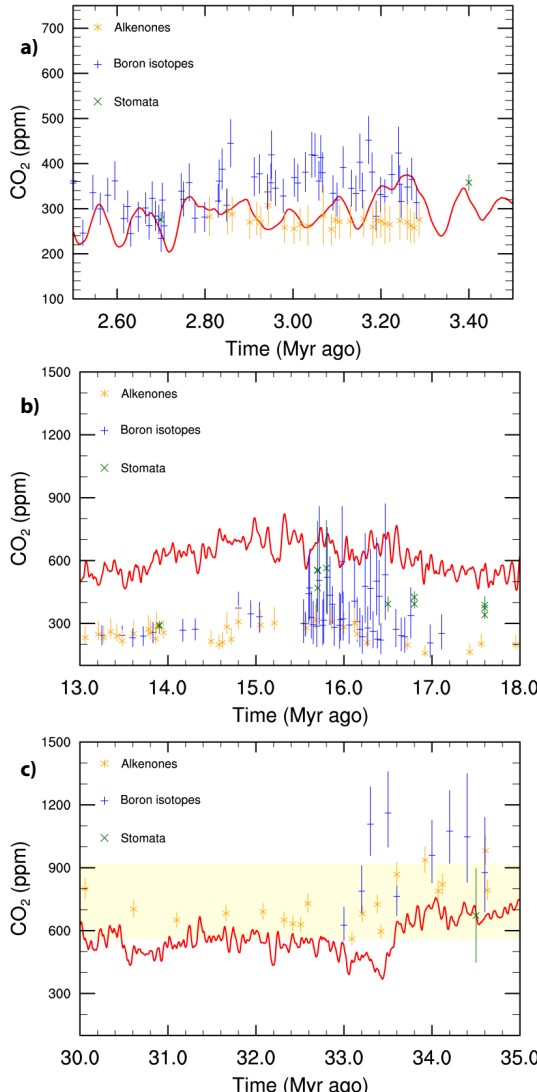

**Figure 5.** Simulated $CO_2$ concentrations of the new reference run (red line) for the (a) Late Pliocene (3.5 to 2.5 Myr ago), (b) Middle Miocene (18 to 13 Myr ago) and **(b)** Early Oligocene (35 to 30 Myr) ago. Shown are 40-kyr running averages. Proxy-data reconstructions based on alkenones (Pagani et al., 1999, 2011; Badger et al., 2013) are indicated by orange asterisks. Boron-isotope-based data (Pearson et al., 2009; Foster et al., 2012; Greenop et al., 2014; Martínez-Botí et al., 2015) are indicated by blue plusses. Stomata-based data (Van der Burgh et al., 1993; Kürschner, 1996; Kürschner et al., 2008; Retallack, 2009) are indicated by green crosses. Yellow shading indicates the range of Antarctic glaciation values found by Gasson et al. (2014).

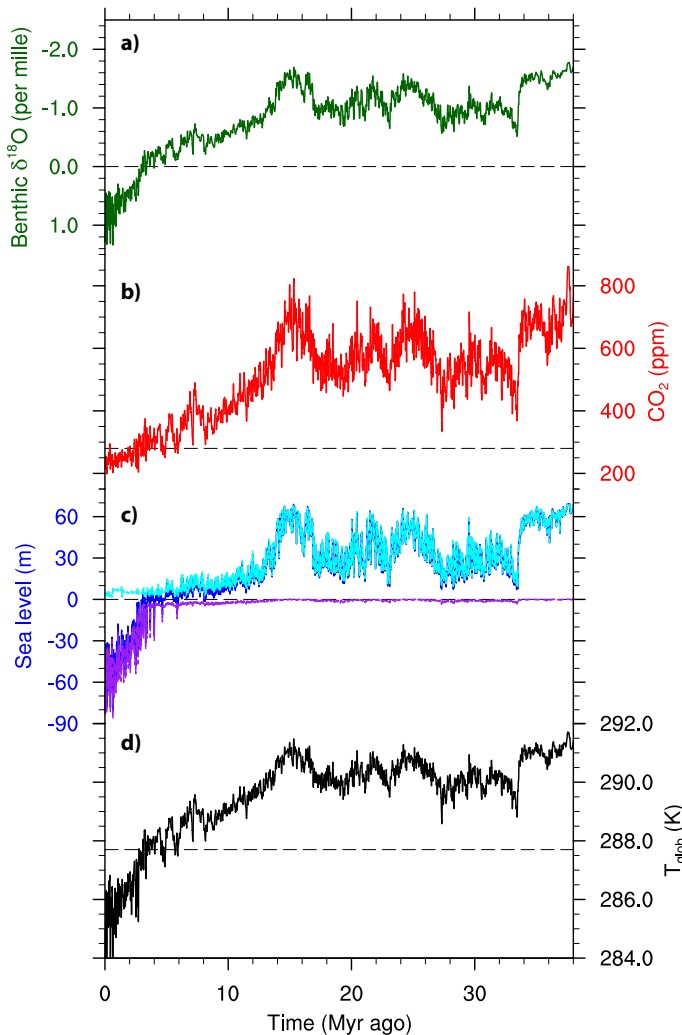

**Figure 6.** Results of the new reference run: (a) benthic $\delta^{18}O$, (b) atmospheric $CO_2$, (c) ice-volume-equivalent sea level in meters above present day (blue), and contributions from the Northern Hemispheric (purple) and Antarctic ice sheets (cyan), (d) global mean temperature ($T_{glob}$). Shown are 40-kyr running averages. Dotted lines represent pre-industrial (PI) values.

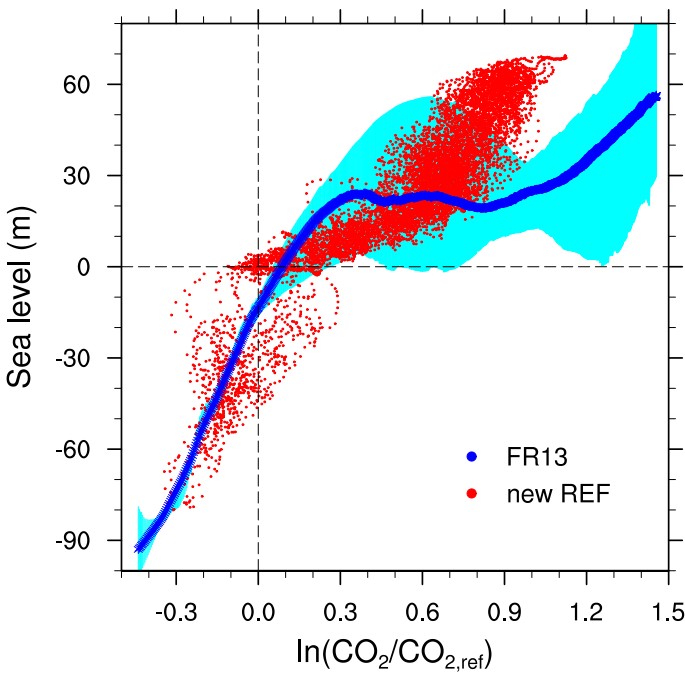

**Figure 7.** Relation between the logarithm of $CO_2$ divided by the PI value of 280 ppm, and ice-volume-equivalent sea level anomalies with respect to PI, for the reference simulation (red dots), compared to the median case of the probabilistic data analysis in Foster and Rohling (2013) (FR13, blue dots) with their 95% uncertainty range in cyan.

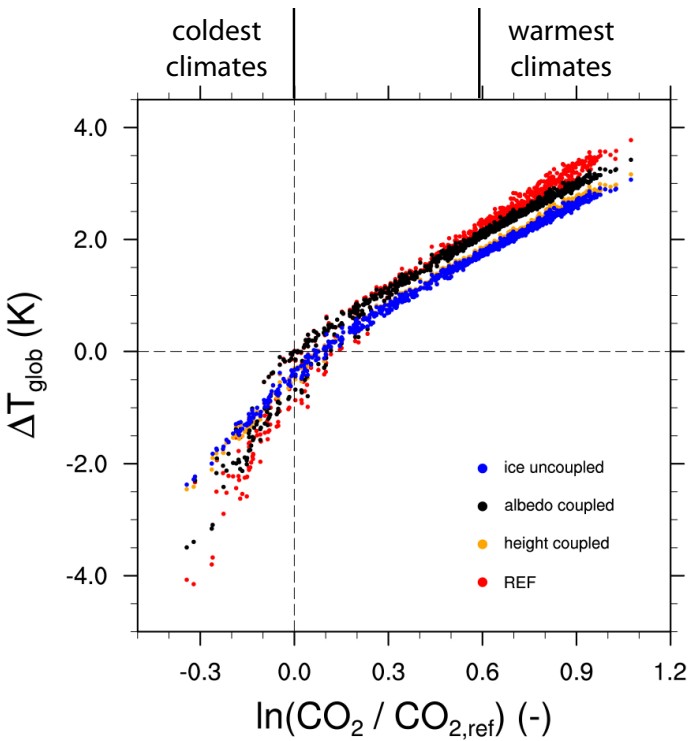

**Figure 8.** Relation between the logarithm of $CO_2$ divided by the PI value of 280 ppm, and global temperature anomalies with respect to PI (of the reference run), for the reference simulation (red dots), the simulation with uncoupled ice (blue dots) and the simulation with only surface height (orange dots) or albedo (black dots) coupled.

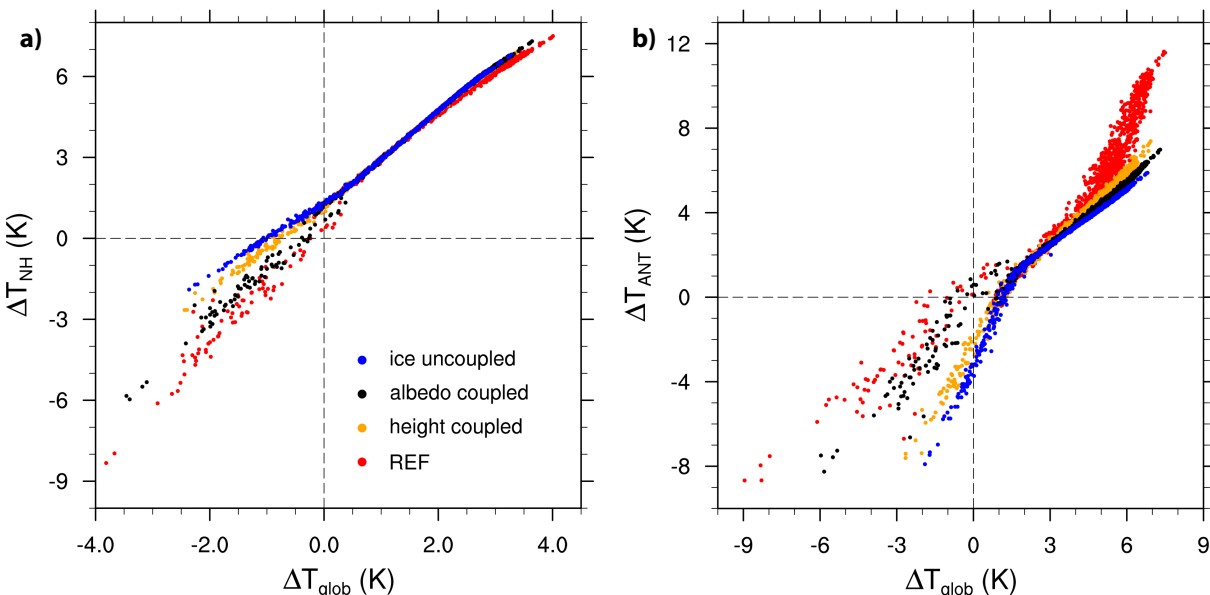

**Figure 9.** Relation between anomalies with respect to PI (of the reference run) of global temperature, and (a) Northern Hemispheric temperature (40 to 80° N), and (b) Antarctic temperature (60 to 90° S), for the reference simulation (red dots), the simulation with uncoupled ice (blue dots) and the simulation with only surface height (orange dots) or albedo (black dots) coupled.

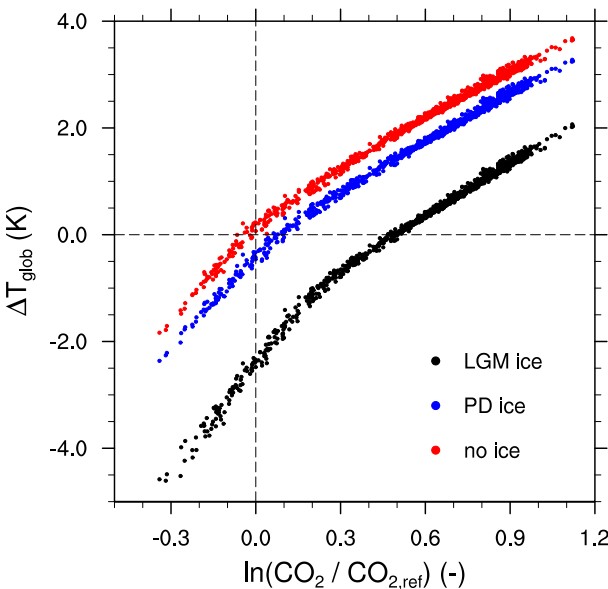

**Figure 10.** Relation between the logarithm of $CO_2$ divided by the PI value of 280 ppm, and global temperature anomalies with respect to PI (of the reference run), for the simulation with ice kept at PD level (blue dots), at LGM level (black dots) and the simulation with all land ice removed (red dots).