# Peer review of "The influence of ice sheets on temperature during the past 38 million years inferred from a one-dimensional ice sheet-climate model"

_Climate of the Past, 2016_

## Referee Comment (RC1) · Anonymous Referee #1 · 19 Jan 2017

General comments:

The authors present results of model simulations of the past 38 million years using a simple zonally averaged energy balance model coupled to a 1D ice sheet model. The presented results contribute to our understanding of climate - ice sheet interactions on very long timescales and therefore the paper represents a valuable contribution to this research field. The use of a relatively simple model is justified by the very long transient simulations which would be too computationally expensive to perform with more complex models. However, in order for the paper to be suitable for publication in Climate of the Past, some minor issues listed below should be addressed.

The model is described only very briefly in the Methodology section. I'm aware that the model is described in more detail in previous publications, but it would be useful to the

reader who is not familiar with the model if some more details would be given (e.g. the resolution of the ice sheet model is not even mentioned in the text). How is the model initialized? Additionally, the way the CO2 concentration is derived in the model and applied as forcing is crucial for the simulations performed and should be described in the paper. I would suggest to at least include the equations for $\delta$18O and CO2.

The description of the experiments used to show the hysteresis behavior of the model is spread over several sections of the paper, which is very confusing to the reader. First it is mentioned in the Methodology section that using different $\delta$18O stacks gives very different results but no reason for that is given until section 3. Then at the end of Section 2 (Page 4, lines 7-15) the hysteresis experiments are described, but it is difficult to understand why these experiments are needed before knowing what the problem is (which is only outlined in Section 3). I would suggest collecting all of this in one section describing the difference between 5Myr and 38Myr simulations, the experiment setup for diagnosing the reason for the differences, the hysteresis behavior and the retuning procedure.

I'm not aware of any other modeling study showing a hysteresis behavior that is caused by the atmosphere model or ocean model when excluding overturning, so it would be interesting to know what is causing this. Because of the relatively short time scale of atmospheric processes, it seems difficult to imagine that the climate model keeps memory of the initial conditions over multimillenial time scales. Could the authors elaborate on this? Are the different hysteresis branches really stable equilibria of the model? Also, does this hysteresis behavior depend on the forcing rate (50 ppm/50 kyr)? What are the initial conditions for these experiments?

The model-derived atmospheric CO2 could be compared with available proxy data (e.g. Beerling and Royer, 2011).

It would be interesting to see also the sea level evolution (maybe also the ice volume evolution separately for NH and Antarctica) and possibly global temperature evolution,

also to make it easier for the reader to interpret Figures 4 and 5.

Figure 3 is very hard to read, especially Figure 3b. Maybe Figure 3b could be split in 3 different plots?

The following sentence in the abstract (Lines 8-9) is not clear, at least not until one has read the rest of the paper: 'Firstly, we investigate the relation between global temperature and CO2, which changes once the model run has experienced high CO2 concentrations.'

---

## Referee Comment (RC2) · Anonymous Referee #2 · 27 Jan 2017

This paper deals with an important issue: the role of ice sheets on the climate evolution since the late Eocene (38 Ma). To achieve this goal, they use simplified climate energy balanced models and also a simplified ice sheet model. Using these tools enables them to simulate very long time spans.

General comment:

Whereas this is an important issue for which there are many unsolved problems as the evolution of Antartica ice-sheets during Oligocene and Miocene and its implication on climate, I feel very uncomfortable with the target, the methodology used and the analysis provided in this paper. These authors had first used this tool to investigate the relationship between cryosphere and climate for 1 million year (Lennert, B Stap, 2014) and extend afterwards to 8 million years (Lennert B Stap, 2016, A). In this new

paper, they enlarge the period to 38 million years. But for many reasons I will explain below, this extension is not convincing with respect to many features: a first obvious one is the role of tectonics on CO2 that the authors perfectly know because they also recently published a paper concerning this issue (Lennert, B Stap, 2016 B). The tectonics, through many different processes, will affect atmospheric pCO2 (see Godderis for a review). For instance opening and closing sea ways may change climate and CO2, orogenesis (E.G Tibetan Plateau Uplift) and plate motion that will impact silicate weathering. Therefore, the extension to 38 Ma they provide in this paper is not really reliable. They reconstruct the pCO2 as a prognostic variable from their model which is indeed important but as they online derive it from radiative perturbation there are missing many fundamental processes. Consequently, their reconstructions of pCO2 over the 38 million years is not in good agreement with data as the authors recognize but instead of accounting for causes of such a disagreement on geological time scale they tuned the model with different parametrization of the clouds physics. This caveat makes the paper not appropriate for publication. Nevertheless, there are potential interesting sensitivity experiments that are possible with such a tool. Another drawback is the fact that they avoid in the introduction to give a context of the state of the art of climate cryosphere interaction using sophisticated GCM as De Conto and Pollard (for instance De Conto and Pollard in Nature 2003, Geoscientific Model Development 2012 and Earth and Planetary Science Letters 2015) developed since many years. One of the major results of De Conto et al. study is to be able to reproduce the evolution of ice sheets since Eocene. They pointed out the importance of cryospheric processes (Pollard and De Conto, EPSL, 2015) that are not discussed at all in this manuscript. Due to these two major problems I don't believe that at this stage such a paper may be published. Nevertheless I will give more details and comments because there is a large room for improvement if the authors want to resubmit their manuscript.

Detailed comments: 1. Abstract First, the relationships between CO2 temperature and ice sheets are consistent within the framework of the modeling study but completely inconsistent with available data concerning CO2 evolution since 38 million years. This

is clearly shown in the paper but not in the abstract itself. Second, the authors insist on very obvious results as for instance it is colder when you get an ice sheet but the most interesting part of the work is to provide many sensitivity experiments. Indeed, this approach, conversely to GCM, as for example De Conto and Pollard (Palaeogeography, Palaeoclimatology, Palaeoecology 2003), allows them to quantify specifically the role of albedo on one side and elevation on the other side. This is not clearly stated in the abstract.

Introduction: This section is a bit short. Some references are missing which may be important. For instance, concerning the Pliocene and Greenland onset, recent publications of Contoux et al (EPSL, 2015) and for MMCO a publication of Hamon (Geology, 2012) constrains on Antarctica ice sheet at MMCO and also Hamon (Climate of the Past, 2013) which depict the role of East Tethys seaway on Antarctica ice sheet 40 million years ago. More importantly, the authors should discuss the interest of their approach compared to the development of GCM studies as those published by De Conto and Pollard (EPSL, 2015) which pinpointed the importance to parametrize the ice sheet with sophisticated models to capture correctly the ice sheet dynamics and therefore to reproduce the ice sheet evolution through Eocene.

Methodology section: First, the authors claimed they used Penthic ïĄďO18 isotope records to infer the temperature of the Ocean, but it is absolutely unclear to me how they really disentangle the part corresponding to ice-sheet melting and the part due to bottom sea surface temperature. This first step has to be clarified, since it is used then to derive through radiative calculation the atmospheric $CO_2$. I strongly believe than in a first step, the authors should have used the different proxy reconstruction used for $CO_2$ as published in the literature, which provides different $CO_2$ evolution (Boron isotopes, Alkenon, leaf stomates,...) to validate their simplified coupled model. Such a strategy based on $CO_2$ reconstruction from data allows to test the response of their tool in terms of cryosphere and climate evolution. Instead, they choose to compute the $CO_2$ from the reconstructed SST, derived from their radiative model. As you know,

there are many reasons and causes that may affect atmospheric CO2, that cannot be accounted for in this very simple modeling tool, especially when dealing with geological time span (38 million years). For instance, seaway changes - and there are many seaway changes in that period (see Zhang et al. Climate of the Past. 2011 ) - or the impact of mountain uplift and associated weathering (see Raymo et al. Nature 1992 and C France-Lanord, Nature, 1997). Therefore, the only processes they captured here, attributing Ocean temperature changes to CO2, is obviously missing a lot of important processes that will change the atmospheric CO2 during that period. Moreover, they use a fixed contribution for the methane in this radiative calculation, (factor 1.3, which is supposed to include the methane radiative perturbation). This value is certainly valid for the last million years, for which data are available, but which is also a very cold period compared to the last 37 million years period they are investigating. Finally, they consider the lapse rate also constant through time whereas, this has been also shown as oversimplified (Svetlana Botsyun et al., Climate of the Past. 2016). These important caveats in the methodology used here, which are absolutely not discussed, imply, as the authors themselves pinpoint, very large underestimation of their computed CO2 when compared to different proxies: the CO2 computed from the temperature record of Zacchos or Raymo, but also those much more accurate and directly obtained from Antarctica ice core (EPICA). The authors claimed that such a mismatch may be overcome by changing the optical properties of the clouds. This is not really serious for me, because it is a kind of tuning without really understanding what is the physics of the problem, but more importantly, they do this tuning for all the time period, whereas there is a strong bias using only EPICA data, which is associated to a very cold period compared to the whole period they are studying. Indeed, most of these 38 million years were much warmer than LGM or present day climate. Therefore, there is no reason for a constant tuning. This also explains why the underestimation is so large for deep time (larger for Zacchos than for Raymo). This methodology by itself induces many problems and leads the authors to explore methodological induced problems, as hysteresis, rather than to really try to capture the dynamics of the cryosphere, or the evolution of

the climate in their result section.

Part 3 results: The part concerning hysteresis is not relevant and convincing for me. Hysteresis has been shown to be an important factor to account for instance in glacial/interglacial cycles (see for instance papers from Paillard Nature 2001, Calov, GRL, 2005. Alvarez-solas Nature Geosci, 2010, De Conto and Pollard Nature 2008...). Here the analyses of the results which depict a strong correlation with the initial climate is not really explained in terms of physics and for me belongs much more to model caveats and development than to the analyses of results interesting enough to be published.

Part 4 discussion: In the discussion section, the summary of the paper is too exhaustive, we really expect a discussion of the results and comparison with the results of other models. For example, these last years, many studies provided by De Conto and Pollard depicted very new results on climate and ice sheets evolution, since the last 40 million years. In this part, we should expect a serious comparison between these results and those provided by the others including the fact that the tools used are different. Therefore, it would be interesting to discuss the result of these two complementary approaches (GCM versus simplified models). Such a discussion will allow the authors to clarify the potential and weaknesses of their method. For instance, simplified tools as used here do not capture important processes that are necessary to simulate ice sheet evolution in GCM. The authors show comment on this point in the discussion section and also highlight on the fact that their tools allow to quantify different forcing factors through the sensitivity experiments.

Conclusion: I strongly believe that there is much room for improvement in this paper. The sections that are devoted to sensitivity experiments (albedo vs topography of the ice sheets) could be a valuable contribution, but at this stage and, accounting for the weaknesses in methodology and construction design of the paper, I think the paper should be rejected. Nevertheless, there are some parts of paper, that, if completely rebuilt could be used and might be a valuable contribution, but in a framework of a

completely new and rethought paper.

DeConto R, Pollard D (2003) Rapid Cenozoic glaciation of Antarctica induced by declining atmospheric CO2. Nature 421:245–249

David Pollard, Robert M. DeConto, Richard B. Alleya. Potential Antarctic Ice Sheet retreat driven by hydrofracturing and ice cliff failure, Earth and Planetary Science Letters, 2015

Robert M. DeContoa, David Pollard. A coupled climate–ice sheet modeling approach to the Early Cenozoic history of the Antarctic ice sheet, Palaeogeography, Palaeoclimatology, Palaeoecology, 2003

Robert M. DeConto, David Pollard, Paul A. Wilson, Heiko Pälike, Caroline H. Lear & Mark Pagani. Thresholds for Cenozoic bipolar glaciation. Nature 455, 652-656 (2 October 2008) | doi:10.1038/nature07337; Received 3 April 2008; Accepted 12 August 2008

David Pollard, Robert M. DeConto. Description of a hybrid ice sheet-shelf model, and application to Antarctica. Geoscientific Model Development; 2012

Svetlana Botsyun, Pierre Sepulchre, Camille Risi and Yannick Donnadieu. Impacts of Tibetan Plateau uplift on atmospheric dynamics and associated precipitation $\delta$18O. Climate of the Past, European Geosciences Union (EGU), 2016, 12 (6), pp.1401-1420. <10.5194/cp-12-1401-2016>

Raymo, M.E and Ruddiman, W.F. Tectonic forcing of late Cenozoic climate. Nature 359, 117-122. 1992 C France-Lanord, LA Derry - Nature, 1997. Organic carbon burial forcing of the carbon cycle from Himalayan erosion

Contoux, C., Dumas, C., Ramstein, G., Jost, A., Dolan, A. M. : Modelling Greenland ice sheet inception and sustainability during the Late Pliocene, Earth and Planet. Sci. Lett., 424, 295-305, doi:10.1016/j.epsl.2015.05.018, 2015

N. Hamon, P. Sepulchre, Y. Donnadieu, A.-J. Henrot, L. François, J.-J. Jaeger and G. Ramstein - "Growth of subtropical forests in Miocene Europe: The roles of carbon dioxide and Antarctic ice volume" - Geology; Jun 2012, Vol. 40 Issue 6, p567

N. Hamon, P. Sepulchre, V. Lefebvre, and G. Ramstein. "The role of eastern Tethys seaway closure in the Middle Miocene Climatic Transition (ca. 14 Ma)". Clim. Past, 9, 2687-2702, 2013

Alvarez-Solas J., Charbit S., Ritz C., Paillard D., Ramstein G., Dumas C., 2010. Links between ocean temperature and iceberg discharge during Heinrich events, Nature Geosci. 3, 122-126.

Reinhard Calov and Andrey Ganopolski. Multistability and hysteresis in the climate-cryosphere system under orbital forcing. GEOPHYSICAL RESEARCH LETTERS, VOL. 32, L21717, doi:10.1029/2005GL024518, 2005

Didier Paillard. Climatology: Glacial hiccups. Nature 409, 147-148 (11 January 2001) | doi:10.1038/35051691

Z. Zhang, K. H. Nisancioglu, F. Flatøy, M. Bentsen, I. Bethke, and H. Wang. Tropical seaways played a more important role than high latitude seaways in Cenozoic cooling. Climate of the Past. 2011. doi:10.5194/cp-7-801-2011

Yves Goddéris, Caroline Roelandt, Jacques Schott, Marie-Claire Pierret, Louis M. François. Towards an Integrated Model of Weathering, Climate, and Biospheric Processes

L.M. François, Y. Goddéris, Isotopic constraints on the Cenozoic evolution of the carbon cycle. Chemical Geology. Volume 145, Issues 3–4, 15 April 1998, Pages 177–212. DOI: 10.2138/rmg.2009.70.9 Published on January 2009, First Published on September 15, 2009

Andrey Ganopolski & Stefan Rahmstorf . Rapid changes of glacial climate simulated in a coupled climate model.Nature 2000.

---

## Author Comment (AC1) · 24 Feb 2017

**REPLY TO THE COMMENTS BY THE REVIEWERS**

Color coding:
Black – comments by reviewers
Blue – reply by authors
Red – textual changes in the manuscript

**General**

Our manuscript has been reviewed by two referees, who have very different opinions. Reviewer #1 is very positive and suggests some minor modifications. Reviewer #2, however, is uncomfortable with a number of points. Most importantly, he/she feels our current model setup lacks important geological processes that might influence the relation between $CO_2$ and climate over time, mainly tectonics and erosion. This is certainly a valid argument and in the revised manuscript we shall express better from the start what the value of our study is in terms of the simulation of $CO_2$ over the past 38 Myr. As requested by this reviewer, we will also improve the introduction by comparing our study to more sophisticated models which are used for particular time slices. We feel that by focusing on the larger picture of transient climate change our current model setup represents a complementary approach to these models, which generally also do not take into account the geological processes mentioned by the reviewer. Furthermore, we will make clearer that our study has a two-fold aim: being a first step in the direction of transient coupled simulations of the climate and cryosphere on long time scales, and quantifying the influence of ice sheets on climate variability. It is an improvement of our earlier research group's earlier work that has been published in several papers which are mentioned in the manuscript. As such, it provides valuable information for the research community. In our opinion, this justifies publication of the current results. Nonetheless, we will take into account as much of the suggestions by both reviewers as possible, which in our opinion will significantly improve the quality of the manuscript. Below, we will answer to the comments of the reviewers.

Structure of the paper
A point where both reviewers agree upon is that the current structure of the paper could be improved. We have therefore decided to follow their suggestions, and structure the revised manuscript in the following manner:

*Introduction*
As requested by reviewer #2, the introduction will be expanded with a discussion of studies using more sophisticated climate and ice sheet models on shorter time scales. This will provide a better perspective of the research field and our contribution to it. We will more clearly state the strengths and weaknesses of our model setup, as well as the purpose of our current study.

*Model*
This section will include a more thorough description of the coupled climate-ice sheet model we use, as well as the inverse routine to simulate $CO_2$. The equations we use to calculate $\delta^{18}O$ and $CO_2$ will be provided as suggested by reviewer #1. The setup of the different model runs we perform will be moved to the Results and Discussion sections to improve readability of the paper.

*Results and Discussion I: Long-term transient simulations*
This section will demonstrate the different $CO_2$ concentrations we obtain over the past 800 kyr when we integrate the model over the past 5 Myr or 38 Myr. We will introduce the hysteresis runs to explore this difference further. Thereafter, we will more clearly describe how and why we re-tune the model. The $CO_2$ will be compared to the proxy data compilation presented in Beerling and Royer (2011), complemented by records published after that study, as requested by reviewer #1.

*Results and Discussion II: Ice sheet-climate interaction*
This section will remain largely the same as Section 4, but including more discussion. As requested by reviewer #1, we will begin this section by presenting the main results of our new 38 Myr reference run: $\delta^{18}O$, $CO_2$, ice-volume-equivalent sea level (total, as well split into contributions from NH and Antarctica) and global mean temperature.

*Summary and Conclusions*
The discussion of the results will be moved to the Results and Discussion sections. This section will only contain a brief summary of our experiments, and the conclusions we derive from them.

**Reviewer #1**

General comments:

The authors present results of model simulations of the past 38 million years using a simple zonally averaged energy balance model coupled to a 1D ice sheet model. The presented results contribute to our understanding of climate - ice sheet interactions on very long timescales and therefore the paper represents a valuable contribution to this research field. The use of a relatively simple model is justified by the very long transient simulations which would be too computationally expensive to perform with more complex models. However, in order for the paper to be suitable for publication in Climate of the Past, some minor issues listed below should be addressed.

We thank the reviewer for considering our work, and we are pleased that he/she agrees with our general approach of using a simplified model to simulate the long-term evolution of climate. We will explain below how we will take the comments into consideration. In our opinion this will improve the quality of the paper, hopefully to the satisfaction of the reviewer.

The model is described only very briefly in the Methodology section. I'm aware that the model is described in more detail in previous publications, but it would be useful to the reader who is not familiar with the model if some more details would be given (e.g. the resolution of the ice sheet model is not even mentioned in the text). How is the model initialized? Additionally, the way the CO2 concentration is derived in the model and applied as forcing is crucial for the simulations performed and should be described in the paper. I would suggest to at least include the equations for $\delta18O$ and CO2.

The reviewer suggests to expand the Methodology section, a point to which reviewer #2 agrees. Therefore, in our new Model section we will include a more thorough explanation of our modelling strategy, including the equations used to calculate $\delta^{18}O$ and $CO_2$.

The description of the experiments used to show the hysteresis behavior of the model is spread over several sections of the paper, which is very confusing to the reader. First it is mentioned in the Methodology section that using different $\delta18O$ stacks gives very different results but no reason for that is given until section 3. Then at the end of Section 2 (Page 4, lines 7-15) the hysteresis experiments are described, but it is difficult to understand why these experiments are needed before knowing what the problem is (which is only outlined in Section 3). I would suggest collecting all of this in one section describing the difference between 5Myr and 38Myr simulations, the experiment setup for diagnosing the reason for the differences, the hysteresis behavior and the retuning procedure.

We will follow the suggestion of the reviewer, and move the description of the model runs to the new Results and Discussion sections. In the new section Results and Discussion I, we will describe the difference between 5 Myr and 38 Myr simulations, the experiment setup for diagnosing the reason for the differences, the hysteresis behavior and the retuning procedure, as the reviewer suggests. This section will end with a comparison of our simulated $CO_2$ to the proxy data compilation presented in Beerling and Royer (2011) and newer records (see also our reply on a further comment by reviewer #1). We will include proper headings marking subsections.

I'm not aware of any other modeling study showing a hysteresis behavior that is caused by the atmosphere model or ocean model when excluding overturning, so it would be interesting to know what is causing this. Because of the relatively short time scale of atmospheric processes, it seems difficult to imagine that the climate model keeps memory of the initial conditions over multimillenial time scales. Could the authors elaborate on this? Are the different hysteresis branches really stable equilibria of the model? Also, does this hysteresis behavior depend on the forcing rate (50 ppm/50 kyr)? What are the initial conditions for these experiments?

The hysteresis runs will be more thoroughly described in the revised manuscript. The questions raised by the reviewer will be addressed: indeed, the different hysteresis branches are stable equilibria of the model. As long as the model is indeed in equilibrium at every time step, the hysteresis behaviour does not depend on the forcing rate: using 50 ppm/100 kyr and 100 ppm/100 kyr leads to the same results. The initial conditions are: 450 ppm $CO_2$, no land ice, glacio-isostatically relaxed present-day topography and present-day insolation.

The model-derived atmospheric CO2 could be compared with available proxy data (e.g. Beerling and Royer, 2011).

The new Results and Discussion I section will contain a comparison of our model results to the proxy data compilation presented in Beerling and Royer (2011) and newer records. Based on this comparison, we will address the caveats and shortcomings of our model. This will clarify the significance of our current model results, as well as indicate a route to go forward from here.

It would be interesting to see also the sea level evolution (maybe also the ice volume evolution separately for NH and Antarctica) and possibly global temperature evolution, also to make it easier for the reader to interpret Figures 4 and 5.

The new Results and Discussion II section will start with a figure showing the main results of our new reference run (after re-tuning): $\delta^{18}O$, $CO_2$, ice-volume-equivalent sea level (total, as well split into contributions from NH and Antarctica) and global mean temperature.

Figure 3 is very hard to read, especially Figure 3b. Maybe Figure 3b could be split in 3 different plots?

To improve readability, Figure 3b will be split as suggested by the reviewer, such that the revised Figure 3 will be composed of 4 subplots (A-D).

The following sentence in the abstract (Lines 8-9) is not clear, at least not until one has read the rest of the paper: 'Firstly, we investigate the relation between global temperature and $_{CO2}$, which changes once the model run has experienced high CO2 concentrations.'

In the revised abstract, we will be clearer on the implication of the analyses of the hysteresis runs:

Firstly, we find that the $CO_2$ simulation over the past 5 Myr is dependent on whether the model run is started at 5 or 38 Myr ago. This is because the relation between $CO_2$ and temperature is subject to hysteresis. When the climate cools from very high $CO_2$ levels, as in the longer 38 Myr run, temperatures in the lower $CO_2$ range of the past 5 Myr are higher than when the climate is initialized at low temperatures. Consequently, the modeled $CO_2$ concentrations are different depending on the initial state.

**Reviewer #2**

This paper deals with an important issue: the role of ice sheets on the climate evolution since the late Eocene (38 Ma). To achieve this goal, they use simplified climate energy balanced models and also a simplified ice sheet model. Using these tools enables them to simulate very long time spans.

General comment:
Whereas this is an important issue for which there are many unsolved problems as the evolution of Antarctica ice-sheets during Oligocene and Miocene and its implication on climate, I feel very uncomfortable with the target, the methodology used and the analysis provided in this paper.

We thank the reviewer for a careful consideration of our work. Unfortunately, he/she is very critical towards our modelling approach. Although the reviewer certainly has some valid points, we still believe our results represent a step forward in our understanding of the influence of ice sheets on long-term climate variability. Below, we will describe why, and

which revisions we make to hopefully ease the objections of the reviewer as much as possible.

These authors had first used this tool to investigate
the relationship between cryosphere and climate for 1 million year (Lennert, B Stap, 2014) and extend afterwards to 8 million years (Lennert B Stap, 2016, A). In this new paper, they enlarge the period to 38 million years. But for many reasons I will explain below, this extension is not convincing with respect to many features: a first obvious one is the role of tectonics on CO2 that the authors perfectly know because they also recently published a paper concerning this issue (Lennert, B Stap, 2016 B). The tectonics, through many different processes, will affect atmospheric pCO2 (see Godderis for a review). For instance opening and closing sea ways may change climate and CO2, orogenesis (E.G Tibetan Plateau Uplift) and plate motion that will impact silicate weathering. Therefore, the extension to 38 Ma they provide in this paper is not really reliable. They reconstruct the pCO2 as a prognostic variable from their model which is indeed important but as they online derive it from radiative perturbation there are missing many fundamental processes. Consequently, their reconstructions of pCO2 over the 38 million years is not in good agreement with data as the authors recognize but instead of accounting for causes of such a disagreement on geological time scale they tuned the model with different parametrization of the clouds physics. This caveat makes the paper not appropriate for publication. Nevertheless, there are potential interesting sensitivity experiments that are possible with such a tool.

The reviewer mentions a number of geological processes that are not taken into account in our model setup, but could influence $CO_2$. However, our study does not concern which processes govern the $CO_2$ concentration in the atmosphere – to address this issue, one would need a carbon cycle model – but what influence $CO_2$ has on the climate, and how ice sheet variability changes this influence. Nevertheless, changing topography could lead to a different relation between $CO_2$ and the coupled climate-ice sheet system, e.g. via changing ocean overturning strength and surface elevation. Indeed, in a previous publication (Stap et al., 2016B) we have explored the effect of the latter process. Our model is unable to simulate some of the aspects shown by proxy data, as we will show in a comparison of our results to the proxy data compilation of Beerling and Royer (2011) and newer records in the revised manuscript. We therefore do not wish to claim that we provide the definitive evolution of $CO_2$ over the past 38 Myr. However, our modelling results clearly represent a step forward from previous studies using a stand-alone ice-sheet model (De Boer et al., 2010), and provide valuable insights into the influence of ice sheet variability on climate. In the revised manuscript, we will be very clear from the start on the purpose of our current study, as well as the caveats that can be addressed in further research.

Another drawback
is the fact that they avoid in the introduction to give a context of the state of the art of climate cryosphere interaction using sophisticated GCM as De Conto and Pollard (for instance De Conto and Pollard in Nature 2003, Geoscientific Model Development 2012 and Earth and Planetary Science Letters 2015) developed since many years. One of the major results of De Conto et al. study is to be able to reproduce the evolution of ice sheets since Eocene. They pointed out the importance of cryospheric processes (Pollard and De Conto, EPSL, 2015) that are not discussed at all in this manuscript.

The second major concern of the reviewer regards the lack of discussion of previous results, in particular the work of Pollard and DeConto in many much-cited publications. This point will be addressed by expanding the introduction of our study to include this discussion. Here, as well as in the new section Results and Discussion II, we will discuss how our results relate to their work, which generally concerns shorter time scales (mostly snap-shot simulations) but using a more sophisticated model setup. We refrain from quantitative comparisons on short time scales, however, since our intention is not to capture any event in great detail, but to provide the larger picture of the long-term influence of ice sheets on the climate. Our results are also not completely independent of the work of Pollard and DeConto, since the inception $CO_2$ level of the Antarctic ice sheet is highly dependent on the parametrisation of the mass balance in our model, and is matched to the one found by Pollard and DeConto (~780 ppm). This will be better explained in the revised manuscript.

Due to these two major problems I don't believe that at this stage such a paper may be published. Nevertheless I will give more details and comments because there is a large room for improvement if the authors want to resubmit their manuscript.

Detailed comments:

1. Abstract First, the relationships between CO2 temperature and ice sheets are consistent within the framework of the modeling study but completely inconsistent with available data concerning CO2 evolution since 38 million years. This is clearly shown in the paper but not in the abstract itself.

We would argue that our results are actually not as bad as the reviewer states here, as we will show in a more rigorous comparison to proxy data in the revised manuscript. However, we will mention the shortcomings of our model and the purpose of our work also in the abstract:

In this study, we use a zonally averaged energy balance climate model bi-directionally coupled to a one-dimensional ice sheet model, capturing the ice-albedo and surface-height-temperature feedbacks. Potentially important transient changes in topographic boundary conditions by tectonics and erosion are not taken into account, but briefly discussed.

Second, the authors insist on very obvious results as for instance it is colder when you get an ice sheet but the most interesting part of the work is to provide many sensitivity experiments. Indeed, this approach, conversely to GCM, as for example De Conto and Pollard (Palaeogeography, Palaeoclimatology, Palaeoecology 2003), allows them to quantify specifically the role of albedo on one side and elevation on the other side. This is not clearly stated in the abstract.

We agree with the reviewer that a main merit of our setup is that it lets us attribute the effect of ice sheets on the climate to two important feedbacks: the ice-albedo feedback and the surface-height-temperature feedback. As this is a main result of this work, it will be mentioned in the revised abstract:

By passing only albedo or surface height changes to the climate model, we can distinguish the separate effects of the ice-albedo and surface-height-temperature feedbacks.

Introduction: This section is a bit short. Some references are missing which may be important. For instance, concerning the Pliocene and Greenland onset, recent publications of Contoux et al (EPSL, 2015) and for MMCO a publication of Hamon (Geology, 2012) constrains on Antarctica ice sheet at MMCO and also Hamon (Climate of the Past, 2013) which depict the role of East Tethys seaway on Antarctica ice sheet 40 million years ago. More importantly, the authors should discuss the interest of their approach compared to the development of GCM studies as those published by De Conto and Pollard (EPSL, 2015) which pinpointed the importance to parametrize the ice sheet with sophisticated models to capture correctly the ice sheet dynamics and therefore to reproduce the ice sheet evolution through Eocene.

We thank the reviewer very much for pointing out these studies. We will expand the Introduction section with a discussion of these papers, which will give the reader a better perspective of the field and our contribution to it. See also our reply to an earlier comment by the reviewer.

Methodology section: First, the authors claimed they used benthic $\delta18O$ isotope records to infer the temperature of the Ocean, but it is absolutely unclear to me how they really disentangle the part corresponding to ice-sheet melting and the part due to bottom sea surface temperature. This first step has to be clarified, since it is used then to derive through radiative calculation the atmospheric CO2. I strongly believe than in a first step, the authors should have used the different proxy reconstruction used for CO2 as published in the literature, which provides different CO2 evolution (Boron isotopes, Alkenon, leaf stomates,. . .) to validate their simplified coupled model. Such a strategy based on CO2 reconstruction from data allows to test the response of their tool in terms of cryosphere and climate evolution. Instead, they choose to compute the CO2 from the reconstructed SST, derived from their radiative model.

The reviewer is unclear as to how our inverse $CO_2$ calculation from benthic $\delta^{18}O$ data works, a concern shared by reviewer #1. Therefore, in our new Model section we will include a thorough explanation of our modelling strategy, including the equations used to calculate $\delta^{18}O$ and $CO_2$. We would like to stress that $CO_2$ is not obtained from SST data, but from benthic $\delta^{18}O$ data which is disentangled into contributions from deep-sea temperature and land ice volume in our model.

In a previous publication (Stap et al., 2014), we have validated our coupled model, using $CO_2$ data from the EPICA Dome C record as input. A reason to refrain from using proxy $CO_2$ data from earlier times as input is that it is currently to scarce and intermittent. Moreover, there is large inter- as well as intra-proxy disagreement. Instead, we opt to use an inverse routine, and compare the results to available proxy data. In the revised manuscript, an explanation of this choice and a comparison to the proxy data compilation of Beerling and Royer (2011) and newer records shall be included.

As you know,
there are many reasons and causes that may affect atmospheric CO2, that cannot be

accounted for in this very simple modeling tool, especially when dealing with geological time span (38 million years). For instance, seaway changes - and there are many seaway changes in that period (see Zhang et al. Climate of the Past. 2011 ) - or the impact of mountain uplift and associated weathering (see Raymo et al. Nature 1992 and C France-Lanord, Nature, 1997). Therefore, the only processes they captured here, attributing Ocean temperature changes to CO2, is obviously missing a lot of important processes that will change the atmospheric CO2 during that period.

We agree with the reviewer that we are missing certain processes (tectonics, erosion) that may affect the relation between $CO_2$ and the coupled climate/cryosphere on the long time scales we investigate, by changing the topographical boundary conditions of the climate model. We therefore do not want to present our $CO_2$ record as the definite simulation of $CO_2$. Rather, as we express in the Discussion/Conclusion section, we pave the way for long time scale simulations, identifying interesting phenomena and potential obstacles. One of these is precisely that these missing processes are important. As this contradicts certain earlier studies (e.g. Foster and Rohling, 2013), we shall make this clearer in our new section Results and Discussion I.

Moreover, they
use a fixed contribution for the methane in this radiative calculation, (factor 1.3, which is supposed to include the methane radiative perturbation). This value is certainly valid for the last million years, for which data are available, but which is also a very cold period compared to the last 37 million years period they are investigating.

We would like to argue that we do not see a better alternative here. The factor 1.3 is indeed derived over the past 800 kyr, the only period over which we have reliable $CH_4$ and $N_2O$ data, as is explained in our publication Stap et al. (2014). We will mention the implication of this modelling choice in the new section Results and Discussion I.

Finally, they
consider the lapse rate also constant through time whereas, this has been also shown as oversimplified (Svetlana Botsyun et al., Climate of the Past. 2016).

Here again, this is the best we can do at this moment. This point will also be included in the discussion.

These important
caveats in the methodology used here, which are absolutely not discussed, imply, as the authors themselves pinpoint, very large underestimation of their computed CO2 when compared to different proxies: the CO2 computed from the temperature record of Zacchos or Raymo, but also those much more accurate and directly obtained from Antarctica ice core (EPICA).

We are afraid that we have not been able to convey our findings well enough to the reviewer. We do not underestimate a proxy computed from the temperature record of Lisiecki and Raymo (2005) and Zachos et al. (2008). The point is that initially we tuned the model to simulate $CO_2$ over the past 800 kyr in agreement with the EPICA Dome C record. Using the same exact model, however, we lose this agreement if we start our model

integration further back in time (38 Myr instead of 5 Myr ago). The disagreement can therefore not be caused by omitted processes, since we in fact use the same model. Hence, we explore the cause of it, by analysing separate hysteresis runs. We will explain this more clearly in the revised manuscript.

The authors claimed that such a mismatch may be overcome by changing the optical properties of the clouds. This is not really serious for me, because it is a kind of tuning without really understanding what is the physics of the problem, but more importantly, they do this tuning for all the time period, whereas there is a strong bias using only EPICA data, which is associated to a very cold period compared to the whole period they are studying. Indeed, most of these 38 million years were much warmer than LGM or present day climate. Therefore, there is no reason for a constant tuning.

We would like to remind the reviewer that all models used to simulate climate and/or ice sheets are to a certain degree tuned in some way. We want to be very clear that the cloud optical thickness is such a tuning factor used in our climate model. Indeed, there is no physical preference for its value before or after re-tuning. Precisely therefore we choose this factor to regain agreement with the EPICA Dome C core. Although we agree that physically cloud properties may change in different climates, we think using a variable parameter setting is cumbersome and does not lead to increased understanding of the studied system. However, we will explain the implications of this choice more precisely in the revised manuscript.

This also explains why the underestimation is so large for deep time (larger for Zacchos than for Raymo). This methodology by itself induces many problems and leads the authors to explore methodological induced problems, as hysteresis, rather than to really try to capture the dynamics of the cryosphere, or the evolution of the climate in their result section.

We are not clear on which underestimation the reviewer refers to here. Also, we are confused by the argument that we do not try to capture the dynamics of the cryosphere, or the evolution of the climate. In our opinion, this is thoroughly dealt with in the Section 4, which will be transformed into a new section Results and Discussion II.

Part 3 results: The part concerning hysteresis is not relevant and convincing for me. Hysteresis has been shown to be an important factor to account for instance in glacial/interglacial cycles (see for instance papers from Paillard Nature 2001, Calov, GRL, 2005. Alvarez-solas Nature Geosci, 2010, De Conto and Pollard Nature 2008. . .). Here the analyses of the results which depict a strong correlation with the initial climate is not really explained in terms of physics and for me belongs much more to model caveats and development than to the analyses of results interesting enough to be published.

We are aware that ice sheet variability, as well as other feedbacks, may cause hysteresis in models and possibly also in the real world. This hysteresis, which is also to some degree contained in our coupled model, however, is inherently different from the hysteresis we explore in this study. We aim to find out why the simulated $CO_2$ over the past 5 Myr is so

different if we start our simulations further back in time. This is in fact not caused by a feedback, but by the core of the model we use, since it shows up even when all feedback processes are shut off. The hysteresis shown by our model may therefore very well be a model caveat. We, however, think this is all the more reason to be honest about its implications, particularly because results of this model have already been published before. Moreover, more sophisticated models have, as far as we know, not been tested for this behaviour. We therefore advise to check for this, before using such models on long-time scales (when computer power permits to do so). This is also in line with our studies objective of identifying interesting phenomena and potential obstacles for transient long-term simulations.

Part 4 discussion: In the discussion section, the summary of the paper is too exhaustive, we really expect a discussion of the results and comparison with the results of other models. For example, these last years, many studies provided by De Conto and Pollard depicted very new results on climate and ice sheets evolution, since the last 40 million years. In this part, we should expect a serious comparison between these results and those provided by the others including the fact that the tools used are different. Therefore, it would be interesting to discuss the result of these two complementary approaches (GCM versus simplified models). Such a discussion will allow the authors to clarify the potential and weaknesses of their method. For instance, simplified tools as used here do not capture important processes that are necessary to simulate ice sheet evolution in GCM. The authors show comment on this point in the discussion section and also highlight on the fact that their tools allow to quantify different forcing factors through the sensitivity experiments.

We realise now that the discussion of our results in relation to previous studies from our own and other research groups is a little bit obscured by the combination with the summary of our current study in this section. We will therefore move the discussion to the two new Results and Discussion sections in the revised manuscript. In the Introduction, and in these Results and Discussion sections, we will also include a discussion of our results with respect to the work of Pollard and DeConto. In the new Summary and Conclusion section we will present our main conclusions.

Conclusion: I strongly believe that there is much room for improvement in this paper. The sections that are devoted to sensitivity experiments (albedo vs topography of the ice sheets) could be a valuable contribution, but at this stage and, accounting for the weaknesses in methodology and construction design of the paper, I think the paper should be rejected. Nevertheless, there are some parts of paper, that, if completely rebuilt could be used and might be a valuable contribution, but in a framework of a completely new and rethought paper.

It is unfortunate that the reviewer thinks the results of our current model setup are not suitable for publication. In our opinion, we provide a complementary approach to snap-shot and short timescale results of more sophisticated models, by focussing on the larger picture of the long-term influence of ice sheets on the climate. We make a step forward from just using stand-alone ice sheet models (e.g. De Boer et al., 2010). Even though this does not lead to any definite answer – there is always a way forward in science – it represents a marked improvement with important results and implications, one of them indeed being the

attribution of the effect of ice sheets on the climate to albedo and topographic changes. We think it is therefore a meaningful contribution to the research field. Nevertheless, as we pointed out, we will make an effort to make the merit of our approach, and the purpose of our study clearer, mainly by improving the structure of the paper and by adding more discussion in order to show the embedding in the existing literature which has more focus on detailed time slice simulations.

**References:**

Beerling, D. J., & Royer, D. L. (2011). Convergent Cenozoic $CO_2$ history. *Nature Geoscience*, *4*(7), 418-420.

De Boer, B., Van de Wal, R. S. W., Bintanja, R., Lourens, L. J., & Tuenter, E. (2010). Cenozoic global ice-volume and temperature simulations with 1-D ice-sheet models forced by benthic $\delta^{18}O$ records. *Annals of Glaciology*, *51*(55), 23-33.

Foster, G. L., & Rohling, E. J. (2013). Relationship between sea level and climate forcing by $CO_2$ on geological timescales. *Proceedings of the National Academy of Sciences*, *110*(4), 1209-1214.

Stap, L. B., Van de Wal, R. S. W., De Boer, B., Bintanja, R., & Lourens, L. J. (2014). Interaction of ice sheets and climate during the past 800 000 years. *Climate of the Past*, *10*(6), 2135.

Stap, L. B., de Boer, B., Ziegler, M., Bintanja, R., Lourens, L. J., & van de Wal, R. S. W. (2016). $CO_2$ over the past 5 million years: Continuous simulation and new $\delta^{11}B$-based proxy data. *Earth and Planetary Science Letters*, *439*, 1-10.

---

## Author Response (AR1)

Date: 26 April 2017
Subject: Resubmission of manuscript

Dear Editor,

Please find attached our manuscript entitled "**The influence of ice sheets on temperature during the past 38 million years**" by L.B. Stap, R.S.W. van de Wal, B. de Boer, R. Bintanja and L.J. Lourens for your consideration for publication in Climate of the Past. It is a significant revision of our earlier submitted manuscript "The influence of ice sheets on the climate during the past 38 million years" (Ref. No. cp-2016-109), which was handled by Dr. Guo.

We believe we have addressed all the concerns raised by the reviewers. In our opinion, this has led to a more comprehensive and more detailed manuscript. It is also better structured, which has increased its readability. This revised manuscript complements a point-by-point answer to the comments of the referees, which was uploaded earlier.

Firstly, the title is revised as you requested, although we feel the new title does not entirely do justice to the content, which consists of results of a coupled ice sheet-climate model. As such, it is more than only about temperature. Secondly, we have expanded the introduction section with more discussion of recent work with coupled ice sheet and climate models. This provides more context to our study. Furthermore, the method section now includes the formulas for calculating benthic $\delta^{18}$O and $CO_2$, so that the reader does not have to go back to older publications to understand our current study. Most importantly, we have included more comparison to earlier work. A section has been added, comparing our $CO_2$ results to proxy data records. Also, the discussion of our temperature records and the interrelationships between CO2, sea level and temperature have been improved, most notably by including a figure comparing our results to the data analyses of Foster and Rohling (2013). Lastly, the conclusion section and abstract now reflect the main merits and findings of this study better.

We hope the new manuscript is to the satisfaction of the reviewers and yourself.

Yours Faithfully,

Lennert Stap

Institute for Marine and Atmospheric research Utrecht (IMAU)
Utrecht University
P.O. Box 80.005
3508 TA Utrecht
The Netherlands
Email: L.B.Stap@uu.nl

**REPLY TO THE COMMENTS BY THE REVIEWERS**

Color coding:
Black – comments by reviewers
Blue – reply by authors

**General**

Our manuscript has been reviewed by two referees, who have very different opinions. Reviewer #1 is very positive and suggests some minor modifications. Reviewer #2, however, is uncomfortable with our study's target, methodology and analyses, and sees no point in its publication. He/she feels our current model setup lacks important geological processes that might influence the relation between $CO_2$ and climate over time. This is certainly a valid argument, that is why we do not wish to claim we present a definite $CO_2$ simulation over the past 38 Myr. We realize this constraint on the implications of our study has not come across well enough before, therefore we will make it clearer in the revised manuscript. Nevertheless, we feel our current model setup represents a complementary approach to more sophisticated models that have been deployed over shorter time scales (mainly snapshot simulations), which generally also do not take into account these geological processes. Furthermore, it is a step forward from using a standalone ice sheet model. Our study has a two-fold aim: being a first step in the direction of transient coupled simulations of the climate and cryosphere on long time scales, and quantifying the influence of ice sheets on climate variability. Therewith it is an intermediate step that provides valuable information for the research community. In our opinion, this justifies publication of the current results. Nonetheless, we will take into account as much of the suggestions by both reviewers as possible, which in our opinion will significantly improve the quality of the manuscript. Below, we will answer to the comments of the reviewers.

A point where both reviewers agree upon is that the current structure of the paper could be improved. We have therefore decided to follow their suggestions, and structure the revised manuscript in the following manner:

*Introduction*
The introduction will be expanded with a discussion of studies using more sophisticated climate and ice sheet models on shorter time scales. This will provide a better perspective of the research field and our contribution to it. We will more clearly state the purpose of our current study.

*Model*
This section will include a more thorough description of the coupled climate-ice sheet model we use, as well as the inverse routine to simulate $CO_2$. The equations we use to calculate $\delta^{18}O$ and $CO_2$ will be provided. The setup of the different model runs we perform will be moved to the Results and Discussion sections to improve readability of the paper.

*Results and Discussion I: Long-term transient simulations*
This section will demonstrate the different $CO_2$ concentrations we obtain over the past 800 kyr when we integrate the model over the past 5 Myr or 38 Myr. We will introduce the hysteresis runs to explore this difference further. Thereafter, we will more clearly describe how and why we re-tune the model. We will present the main results of our new 38 Myr reference run: $\delta^{18}O$, $CO_2$, ice-volume-equivalent sea level (total, as well split into contributions from NH and Antarctica) and global mean temperature. The $CO_2$ will be compared to the proxy data by Beerling and Royer (2011), and based on this, we will address the caveats and shortcomings of our model, and thereby the scope and target of our study.

*Results and Discussion II: Ice sheet-climate interaction*
This section will remain largely the same as Section 4, but including more discussion.

*Summary and Conclusions*
The discussion of the results will be moved to the Results and Discussion sections. This section will only contain a brief summary of our experiments, and the conclusions we derive from them.

**Reviewer #1**

General comments:

The authors present results of model simulations of the past 38 million years using a simple zonally averaged energy balance model coupled to a 1D ice sheet model. The presented results contribute to our understanding of climate - ice sheet interactions on very long timescales and therefore the paper represents a valuable contribution to this research field. The use of a relatively simple model is justified by the very long transient simulations which would be too computationally expensive to perform with more complex models. However, in order for the paper to be suitable for publication in Climate of the Past, some minor issues listed below should be addressed.

We thank the reviewer for considering our work, and we are pleased that he/she agrees with our general approach. We will explain below how we will take the comments into consideration. In our opinion this will improve the quality of the paper, hopefully to the satisfaction of the reviewer.

The model is described only very briefly in the Methodology section. I'm aware that the model is described in more detail in previous publications, but it would be useful to the reader who is not familiar with the model if some more details would be given (e.g. the resolution of the ice sheet model is not even mentioned in the text). How is the model initialized? Additionally, the way the CO2 concentration is derived in the model and applied as forcing is crucial for the simulations performed and should be described in the paper. I would suggest to at least include the equations for $\delta18O$ and CO2.

The reviewer suggests to expand the Methodology section, a point to which reviewer #2 agrees. Therefore, in our new Model section we will include a more thorough explanation of our modelling strategy, including the equations used to calculate $\delta^{18}O$ and $CO_2$.

The description of the experiments used to show the hysteresis behavior of the model is spread over several sections of the paper, which is very confusing to the reader. First it is mentioned in the Methodology section that using different $\delta18O$ stacks gives very different results but no reason for that is given until section 3. Then at the end of Section 2 (Page 4, lines 7-15) the hysteresis experiments are described, but it is difficult to understand why these experiments are needed before knowing what the problem is (which is only outlined in Section 3). I would suggest collecting all of this in one section describing the difference between 5Myr and 38Myr simulations, the experiment setup for diagnosing the reason for the differences, the hysteresis behavior and the retuning procedure.

We will follow the suggestion of the reviewer, and move the description of the model runs to the new Results and Discussion sections. In the new section Results and Discussion I, we will describe the difference between 5 Myr and 38 Myr simulations, the experiment setup for diagnosing the reason for the differences, the hysteresis behavior and the retuning procedure, as the reviewer suggests. This section will end with a comparison of our simulated $CO_2$ to the proxy data of Beerling and Royer (2011) (see also our reply on a further comment by reviewer #1).

I'm not aware of any other modeling study showing a hysteresis behavior that is caused by the atmosphere model or ocean model when excluding overturning, so it would be interesting to know what is causing this. Because of the relatively short time scale of atmospheric processes, it seems difficult to imagine that the climate model keeps memory of the initial conditions over multimillenial time scales. Could the authors elaborate on this? Are the different hysteresis branches really stable equilibria of the model? Also, does this hysteresis behavior depend on the forcing rate (50 ppm/50 kyr)? What are the initial conditions for these experiments?

The hysteresis runs will be more thoroughly described in the revised manuscript. The questions raised by the reviewer will be addressed.

The model-derived atmospheric CO2 could be compared with available proxy data (e.g. Beerling and Royer, 2011).

The new Results and Discussion I section will contain a comparison of our model results to the proxy data of Beerling and Royer (2011). Based on this comparison, we will address the caveats and shortcomings of our model. This will clarify the purport of our current model results, as well as indicate a route to go forward from here.

It would be interesting to see also the sea level evolution (maybe also the ice volume evolution separately for NH and Antarctica) and possibly global temperature evolution, also to make it easier for the reader to interpret Figures 4 and 5.

The new Results and Discussion II section will start with a figure showing the main results of our new reference run (after re-tuning): $\delta^{18}O$, $CO_2$, ice-volume-equivalent sea level (total, as well split into contributions from NH and Antarctica) and global mean temperature.

Figure 3 is very hard to read, especially Figure 3b. Maybe Figure 3b could be split in 3 different plots?

To improve readability, Figure 3b will be split as suggested by the reviewer, such that the revised Figure 3 will be composed of 4 subplots (A-D).

The following sentence in the abstract (Lines 8-9) is not clear, at least not until one has read the rest of the paper: 'Firstly, we investigate the relation between global temperature and $CO_2$, which changes once the model run has experienced high CO2 concentrations.'

In the revised abstract, we will be clearer on the implication of the analyses of the hysteresis runs.

**Reviewer #2**

This paper deals with an important issue: the role of ice sheets on the climate evolution since the late Eocene (38 Ma). To achieve this goal, they use simplified climate energy balanced models and also a simplified ice sheet model. Using these tools enables them to simulate very long time spans.

General comment:
Whereas this is an important issue for which there are many unsolved problems as the evolution of Antartica ice-sheets during Oligocene and Miocene and its implication on climate, I feel very uncomfortable with the target, the methodology used and the analysis provided in this paper.

We thank the reviewer for a careful consideration of our work. Unfortunately, he/she is very critical towards our modelling approach. Although the reviewer certainly has some valid points, we still believe our results represent a step forward in our understanding of the influence of ice sheets on long-term climate variability. Below, we will describe why, and which revisions we make to hopefully ease the objections of the reviewer as much as possible.

These authors had first used this tool to investigate the relationship between cryosphere and climate for 1 million year (Lennert, B Stap, 2014) and extend afterwards to 8 million years (Lennert B Stap, 2016, A). In this new paper, they enlarge the period to 38 million years. But for many reasons I will explain below, this extension is not convincing with respect to many features: a first obvious one is the role of tectonics on CO2 that the authors perfectly know because they also recently published a paper concerning this issue (Lennert, B Stap, 2016 B). The tectonics, through many different processes, will affect atmospheric pCO2 (see Godderis for a review). For instance opening and closing sea ways may change climate and CO2, orogenesis (E.G Tibetan Plateau Uplift) and plate motion that will impact silicate weathering. Therefore, the extension to 38 Ma they provide in this paper is not really reliable. They reconstruct the pCO2 as a prognostic variable from their model which is indeed important but as they online derive it from radiative perturbation there are missing many fundamental processes. Consequently, their reconstructions of pCO2

over the 38 million years is not in good agreement with data as the authors recognize but instead of accounting for causes of such a disagreement on geological time scale they tuned the model with different parametrization of the clouds physics. This caveat makes the paper not appropriate for publication. Nevertheless, there are potential interesting sensitivity experiments that are possible with such a tool.

The reviewer mentions a number of geological processes that are not taken into account in our model setup, but could influence $CO_2$. However, our study does not concern which processes govern the $CO_2$ concentration in the atmosphere – to address this issue, one would need a carbon cycle model – but what influence $CO_2$ has on the climate, and how ice sheet variability changes this influence. Nevertheless, changing topography could lead to a different relation between $CO_2$ and the coupled climate-ice sheet system, e.g. via changing ocean overturning strength and surface elevation. Indeed, in a previous publication (Stap et al., 2016B) we have explored the effect of the latter process. Our model is unable to simulate some of the aspects shown by proxy data, as we will show in a comparison of our results to the data of Beerling and Royer (2011) in the revised manuscript. We therefore do not wish to claim that we provide the final answer to the evolution of $CO_2$ over the past 38 Myr. However, our modelling results clearly represent a step forward from previous studies using a stand-alone ice-sheet model (De Boer et al., 2010), and provide valuable insights into the influence of ice sheet variability on climate. In the revised manuscript, we will be very clear on the purpose of our current study, as well as the caveats that can be addressed in further research.

Another drawback
is the fact that they avoid in the introduction to give a context of the state of the art of climate cryosphere interaction using sophisticated GCM as De Conto and Pollard (for instance De Conto and Pollard in Nature 2003, Geoscientific Model Development 2012 and Earth and Planetary Science Letters 2015) developed since many years. One of the major results of De Conto et al. study is to be able to reproduce the evolution of ice sheets since Eocene. They pointed out the importance of cryospheric processes (Pollard and De Conto, EPSL, 2015) that are not discussed at all in this manuscript.

The second mayor concern of the reviewer regards the lack of discussion of previous results, in particular the work of Pollard and DeConto in many much-cited publications. This point is easily addressed by expanding the introduction of our study to include this discussion. Here, as well as in the new section Results and Discussion II, we will discuss how our results relate to their work, which generally concerns shorter time scales (mostly snap-shot simulations) but using a more sophisticated model setup. We refrain from quantitative comparisons on short time scales, however, since our intention is not to capture any event in great detail, but to provide the larger picture of the long-term influence of ice sheets on the climate. Our results are also not completely independent of the work of Pollard and DeConto, since the inception $CO_2$ level of the Antarctic ice sheet is highly dependent on the parametrisation of the mass balance in our model, and is matched to the one found by Pollard and DeConto (~780 ppm). This will be explained in the revised manuscript.

Due to these two major problems I don't believe that at this stage such a paper may be published. Nevertheless I will give more details and comments because there is a large room for improvement if the authors want to resubmit their manuscript.

Detailed comments:

1. Abstract First, the relationships between CO2 temperature and
ice sheets are consistent within the framework of the modeling study but completely
inconsistent with available data concerning CO2 evolution since 38 million years. This
is clearly shown in the paper but not in the abstract itself.

We would argue that our results are actually not as bad as the reviewer states here, as we
will show in a more rigorous comparison to proxy data in the revised manuscript. However,
we will mention the shortcomings of our model also in the abstract.

Second, the authors insist
on very obvious results as for instance it is colder when you get an ice sheet but the
most interesting part of the work is to provide many sensitivity experiments. Indeed, this
approach, conversely to GCM, as for example De Conto and Pollard (Palaeogeography,
Palaeoclimatology, Palaeoecology 2003), allows them to quantify specifically the role
of albedo on one side and elevation on the other side. This is not clearly stated in the
abstract.

We agree with the reviewer that a main merit of our setup is that it lets us attribute the
effect of ice sheets on the climate to two important feedbacks: the ice-albedo feedback and
the surface-height-temperature feedback. As this is a main result of this work, it will be
mentioned in the revised abstract.

Introduction: This section is a bit short. Some references are missing which may
be important. For instance, concerning the Pliocene and Greenland onset, recent
publications of Contoux et al (EPSL, 2015) and for MMCO a publication of Hamon
(Geology, 2012) constrains on Antarctica ice sheet at MMCO and also Hamon (Climate
of the Past, 2013) which depict the role of East Tethys seaway on Antarctica ice sheet
40 million years ago. More importantly, the authors should discuss the interest of their
approach compared to the development of GCM studies as those published by De
Conto and Pollard (EPSL, 2015) which pinpointed the importance to parametrize the
ice sheet with sophisticated models to capture correctly the ice sheet dynamics and
therefore to reproduce the ice sheet evolution through Eocene.

We thank the reviewer very much for pointing our attention at these studies. We will
expand the Introduction section with a discussion of these works, which will give the reader
a better perspective of the field and our contribution to it. See also our reply to an earlier
comment by the reviewer.

Methodology section: First, the authors claimed they used Penthic ïA¸d'O18 isotope
records to infer the temperature of the Ocean, but it is absolutely unclear to me how
they really disentangle the part corresponding to ice-sheet melting and the part due to
bottom sea surface temperature. This first step has to be clarified, since it is used then
to derive through radiative calculation the atmospheric CO2. I strongly believe than
in a first step, the authors should have used the different proxy reconstruction used
for CO2 as published in the literature, which provides different CO2 evolution (Boron

isotopes, Alkenon, leaf stomates,. . .) to validate their simplified coupled model. Such a strategy based on CO2 reconstruction from data allows to test the response of their tool in terms of cryosphere and climate evolution. Instead, they choose to compute the CO2 from the reconstructed SST, derived from their radiative model.

The reviewer is unclear as to how our inverse $CO_2$ calculation from benthic $\delta^{18}O$ data works, a concern shared by reviewer #1. Therefore, in our new Model section we will include a more thorough explanation of our modelling strategy, including the equations used to calculate $\delta^{18}O$ and $CO_2$. We would like to stress that $CO_2$ is not obtained from SST data, but from benthic $\delta^{18}O$ data which is disentangled into contributions from deep-sea temperature and land ice volume in our model.

In a previous publication (Stap et al., 2014), we have validated our coupled model, using $CO_2$ data from the EPICA Dome C record as input. A reason to refrain from using proxy $CO_2$ data from earlier times as input is that it is currently to scarce and intermittent. Moreover, there is large inter- as well as intra-proxy disagreement. Instead, we opt to use an inverse routine, and compare the results to available proxy data. In the revised manuscript, an explanation of this choice and a comparison to the data of Beerling and Royer (2011) shall be included.

As you know,
there are many reasons and causes that may affect atmospheric CO2, that cannot be accounted for in this very simple modeling tool, especially when dealing with geological time span (38 million years). For instance, seaway changes - and there are many seaway changes in that period (see Zhang et al. Climate of the Past. 2011 ) - or the impact of mountain uplift and associated weathering (see Raymo et al. Nature 1992 and C France-Lanord, Nature, 1997). Therefore, the only processes they captured here, attributing Ocean temperature changes to CO2, is obviously missing a lot of important processes that will change the atmospheric CO2 during that period.

We agree with the reviewer that we are missing certain processes that affect the relation between $CO_2$ and the coupled climate/cryosphere on these long time scales. We therefore do not want to present our $CO_2$ record as the definite simulation of $CO_2$. Rather, as we express in the Discussion/Conclusion section, we pave the way for long time scale simulations, identifying interesting phenomena and potential obstacles. One of these is precisely that the processes mentioned by the reviewer are important. As this contradicts certain earlier studies (e.g. Foster and Rohling, 2013), we shall make this clearer in our new section Results and Discussion I.

Moreover, they
use a fixed contribution for the methane in this radiative calculation, (factor 1.3, which is supposed to include the methane radiative perturbation). This value is certainly valid for the last million years, for which data are available, but which is also a very cold period compared to the last 37 million years period they are investigating.

We would like to argue that we do not see a better alternative here. The factor 1.3 is indeed derived over the past 800 kyr, the only period over which we have reliable CH4 and N2O data, as is explained in our publication Stap et al. (2014). We will mention the implication of this modelling choice in the new section Results and Discussion I.

Finally, they
consider the lapse rate also constant through time whereas, this has been also shown
as oversimplified (Svetlana Botsyun et al., Climate of the Past. 2016).

Here again, this is the best we can do at this moment. This point will also be included in the discussion.

These important
caveats in the methodology used here, which are absolutely not discussed, imply, as
the authors themselves pinpoint, very large underestimation of their computed CO2
when compared to different proxies: the CO2 computed from the temperature record
of Zacchos or Raymo, but also those much more accurate and directly obtained from
Antarctica ice core (EPICA).

We are afraid that we have not been able to convey our findings well enough to the reviewer. We do not underestimate a proxy computed from the temperature record of Raymo and Zacchos. The point is that initially we tuned the model to simulate $CO_2$ over the past 800 kyr in agreement with the EPICA Dome C record. Using the same exact model, however, we lose this agreement if we start our model integration further back in time (38 Myr instead of 5 Myr ago). The disagreement can therefore not be caused by omitted processes, since we in fact use the same model. Hence, we explore the cause of it, by analysing separate hysteresis runs. We will explain this more clearly in the revised manuscript.

The authors claimed that such a mismatch may be overcome
by changing the optical properties of the clouds. This is not really serious for
me, because it is a kind of tuning without really understanding what is the physics of
the problem, but more importantly, they do this tuning for all the time period, whereas
there is a strong bias using only EPICA data, which is associated to a very cold period
compared to the whole period they are studying. Indeed, most of these 38 million years
were much warmer than LGM or present day climate. Therefore, there is no reason for
a constant tuning.

We would like to remind the reviewer that all models used to simulate climate and/or ice sheets are to lesser or stronger degree tuned in some way. We want to be very clear that the cloud optical thickness is such a tuning factor used in our climate model. Indeed, there is no physical preference for its value before or after re-tuning. Precisely therefore we choose this factor to regain agreement with the EPICA Dome C core. Although we agree that physically cloud properties may change in different climates, we think using a variable parameter setting is cumbersome and does not lead to increased understanding of the studied system. However, we will explain the implications of this choice more precisely in the revised manuscript.

This also explains why the underestimation is so large for deep time
(larger for Zacchos than for Raymo). This methodology by itself induces many problems
and leads the authors to explore methodological induced problems, as hysteresis,

rather than to really try to capture the dynamics of the cryosphere, or the evolution of the climate in their result section.

We are not clear on which underestimation the reviewer refers to here. Also, we are confused by the argument that we do not try to capture the dynamics of the cryosphere, or the evolution of the climate. In our opinion, this is thoroughly dealt with in the Section 4, which will be transformed into a new section Results and Discussion II.

Part 3 results: The part concerning hysteresis is not relevant and convincing for me. Hysteresis has been shown to be an important factor to account for instance in glacial/interglacial cycles (see for instance papers from Paillard Nature 2001, Calov, GRL, 2005. Alvarez-solas Nature Geosci, 2010, De Conto and Pollard Nature 2008. . .). Here the analyses of the results which depict a strong correlation with the initial climate is not really explained in terms of physics and for me belongs much more to model caveats and development than to the analyses of results interesting enough to be published.

We realise that ice sheet variability, as well as other feedbacks, may cause hysteresis in models and possibly also in the real world. This hysteresis, which is also to some degree contained in our coupled model, however, is inherently different from the hysteresis we explore in this study. We aim to find out why the simulated $CO_2$ over the past 5 Myr is so different if we start our simulations further back in time. This is in fact not caused by a feedback, but by the core of the model we use, since it shows up even when all feedback processes are shut off. The hysteresis shown by our model may therefore very well be a model caveat. We, however, think this is all the more reason to be honest about its implications and not sweep it under the carpet, particularly because results of this model have already been published before. Moreover, more sophisticated models have, as far as we know, not been tested for this behaviour. We therefore advise to check for this, before using such models on long-time scales (when computer power permits to do so). This is also in line with our studies objective of identifying interesting phenomena and potential obstacles for transient long-term simulations.

Part 4 discussion: In the discussion section, the summary of the paper is too exhaustive, we really expect a discussion of the results and comparison with the results of other models. For example, these last years, many studies provided by De Conto and Pollard depicted very new results on climate and ice sheets evolution, since the last 40 million years. In this part, we should expect a serious comparison between these results and those provided by the others including the fact that the tools used are different. Therefore, it would be interesting to discuss the result of these two complementary approaches (GCM versus simplified models). Such a discussion will allow the authors to clarify the potential and weaknesses of their method. For instance, simplified tools as used here do not capture important processes that are necessary to simulate ice sheet evolution in GCM. The authors show comment on this point in the discussion section and also highlight on the fact that their tools allow to quantify different forcing factors through the sensitivity experiments.

We realise now that the discussion of our results in relation to previous studies from our own and other research groups is a little bit obscured by the combination with the summary

of our current study in this section. We will therefore move the discussion to the two new Results and Discussion sections in the revised manuscript. In the introduction, and in these Results and Discussion sections, we will also include a discussion of our results with respect to the work of Pollard and DeConto. In the new Summary and Conclusion section we will present our main conclusions.

Conclusion: I strongly believe that there is much room for improvement in this paper. The sections that are devoted to sensitivity experiments (albedo vs topography of the ice sheets) could be a valuable contribution, but at this stage and, accounting for the weaknesses in methodology and construction design of the paper, I think the paper should be rejected. Nevertheless, there are some parts of paper, that, if completely rebuilt could be used and might be a valuable contribution, but in a framework of a completely new and rethought paper.

It is unfortunate that the reviewer thinks the results of our current model setup are not suitable for publication. In our opinion, we provide a complementary approach to snap-shot and short timescale results of more sophisticated models, and make a step forward from just using stand-alone ice sheet models (e.g. De Boer et al., 2010). Even though this does not lead to any definite answer – there is always a way forward in science – it represents a marked improvement with important results and implications, one of them indeed being the attribution of the effect of ice sheets on the climate to albedo and topographic changes. We think it is therefore a meaningful contribution to the research field. Nevertheless, as we pointed out, we will make an effort to make the merit of our approach, and the purpose of our study clearer, mainly by improving the structure of the paper and by adding more discussion.

**References:**

[revised manuscript text omitted]

---

## Referee Report (RR1)

**Reviews on "The influence of ice sheets on temperature during the past 38 million years" by Stap et al.**

Comments

This paper carried out transient simulations over the past 38 millions of years, an ice-sheet model coupled with an energy balance model. This work aims at studying how ice-sheets impact on climate changes in the past 38 million years.

Although the model is simple and idealized, this work is one of very few integrations over such a long time. The study is a good reference for future studies with mode sophisticate models. I noticed that it has taken a quite long period in paper reviewing and revising. I found that the authors have made sufficient changes in the revised version. Thus, I would recommend publication of the paper.

I read through the manuscript and did not find major problems.

There is one minor thing: page 11, line 7, "haven " –> "have".

---

## Author Response (AR2)

Date: 21 August 2017
Subject: Resubmission of manuscript

Dear Editor,

Please find attached our manuscript entitled "**The influence of ice sheets on temperature during the past 38 Myr inferred from a one-dimensional ice sheet-climate model**" by L.B. Stap, R.S.W. van de Wal, B. de Boer, R. Bintanja and L.J. Lourens for your consideration for publication in Climate of the Past. It is a revision of our earlier submitted manuscript "The influence of ice sheets on temperature during the past 38 million years" (Ref. No. cp-2016-109), which was handled by Dr. Guo.

We believe we have addressed all remaining concerns of the four reviewers. We have revised the manuscript according to their suggestions. Most importantly, we have expanded the section on the model hysteresis with an analysis of the energy balance. This includes three new figures, two of which we have placed in the supplementary material. In our opinion, the new analysis has made this section more comprehensive, and therefore better understandable and more convincing.

We hope the new manuscript is to the satisfaction of the reviewers and yourself.

Yours Faithfully,

Lennert Stap

Institute for Marine and Atmospheric research Utrecht (IMAU)
Utrecht University
P.O. Box 80.005
3508 TA Utrecht
The Netherlands
Email: L.B.Stap@uu.nl

REPLY TO THE COMMENTS BY THE REVIEWERS

Color coding:
Black – comments by reviewers
Blue – reply by authors
Red – Changes made in the manuscript (page and line numbers refer to the revised manuscript)

**Reviewer #1**
In the revised version of the paper, the authors have addressed most of my comments. However, they have not properly addressed one of my concerns:

'I'm not aware of any other modeling study showing a hysteresis behavior that is caused by the atmosphere model or ocean model when excluding overturning, so it would be interesting to know what is causing this. Because of the relatively short time scale of atmospheric processes, it seems difficult to imagine that the climate model keeps memory of the initial conditions over multimillenial time scales. Could the authors elaborate on this?'

An explanation of what processes are causing the hysteresis when ice sheets and CO2 are prescribed still remains unclear to me. Since this hysteresis makes a difference of ~50-100 ppm in modeled atmospheric CO2 over the last 10 Myr (Fig.2), it seems important to me that the physical mechanism behind this behavior is explained. For instance, what is different between 'up' and 'down' simulations at 150 ppm in the atmosphere and/or ocean? Understanding this behavior is important also to exclude the presence of problems in the modeling setup.

We thank the reviewer for considering our manuscript once more. We have addressed his/her remaining concern by adding a new figure (Figure 4) showing the difference in the meridional atmospheric temperature distribution at 150 ppm between the up and down runs. We have chosen this quantity, because we focus on this in later sections. The difference in oceanic temperature, however, is similar. We have also added an analysis (including two new supplementary figures) of the energy balance calculations to section 3.1, by showing the energy imbalance at the top of the atmosphere in both the inverse and the forward runs of the model. The energy imbalance shows the same branches as global temperature when plotted against $CO_2$. This is also the case in a model run, where surface albedo and ocean overturning strength are kept at present-day configuration. This leads us to conclude that the convergence towards different imbalances at the same $CO_2$, which is most likely a consequence of the temperature dependency of atmospheric transparency to short and long wave radiation, is at the heart of the hysteresis in the $CO_2$-temperature relation that we find. We believe the extra analysis has made the paper more comprehensive and therefore clearer, hopefully to the satisfaction of the reviewer.

*Page 6, line 9-10:*
The difference is generally larger at higher than at lower latitudes, peaking at $60^0$ S and $80^0$ N (Fig. 4, red line).

*Page 6, line 15-33:*
This difference is now more uniformly spread over the globe, albeit slightly larger in the Southern than in the Northern Hemisphere (Fig. 4, blue line). Having ruled out albedo and ocean overturning strength, which only act to enhance it, the source of the hysteresis is most likely the temperature dependency of atmospheric transparency to short and long wave radiation in the energy balance calculations (Bintanja, 1996). Unfortunately, disabling this dependency in the energy balance calculations leads to numerical instability, causing the model to crash shortly after initiation. However, its influence is suggested by the model converging to a smaller steady-state energy imbalance at the top of the atmosphere, when the model has previously experienced high $CO_2$ values. This is the case in both the fully coupled run (Suppl. Fig. 1a) and the run where surface albedo and ocean overturning strength are kept fixed (Suppl. Fig. 1b). An energy imbalance at the top of the atmosphere when the atmospheric and oceanic temperatures have equilibrated, means there is an energy leak. When this energy leak is positive, the outgoing radiation and therefore the radiative temperature of the Earth are underestimated. For steady-state PI conditions, most General Circulation Models (GCMs) show positive energy imbalances, typically below 2 $Wm^{-2}$ (Lucarini and Ragone, 2011). In our model, more energy is lost at higher $CO_2$ concentrations, meaning the model simulates increasingly too cold temperatures towards warmer climates. Therefore, the climate sensitivity is underestimated. However, the energy leak remains relatively small: 0 to 4.6 $Wm^{-2}$ globally averaged for $CO_2$ values between 150 and 1200 ppm (Suppl. Fig. 1a). In the inverse mode runs, the energy imbalance is indeed also different: the energy imbalance is smaller in the 38-Myr run than in the 5-Myr run (Suppl. Fig. 2, blue and green lines), indicating that the modelled climate states in these runs are in different branches of the curve shown in Suppl. Fig. 1. Although the difference in energy imbalance is small between the different branches, it evidently has important implications for the simulated $CO_2$ in inverse mode.

*Page 8, line 1-2:*
Furthermore, the energy imbalance at the top of the atmosphere is not affected significantly by retuning the model, and remains lower in this simulation than in the 5-Myr run (Suppl. Fig. 2, red line).

**Reviewer #2**
Review of revised manuscript cp-2016-109 "The influence of ice sheets on temperature during the past 38 million years "

After reading of the authors' response and the revised manuscript, in my view there are some issues with that manuscript that still remain, which warrant a minor revision before it is suitable for publication.

General comments:

The authors' effort to address the first set of comments from the reviewers is appreciated, and in my opinion the revisions have improved the manuscript. The study should aid in the understanding and interpretation of the long term climate - ice sheet interaction in the transient simulations, which also sets a reference simulation for more sophisticated models. The scope of the new version of the paper, as the revised title and abstract state, is properly

addressed. The overall structure of the paper is substantially improved. The introduction now includes useful discussion of complex climate model simulations on shorter time scales. The equations of the model are added so the paper is easier to read.

Sec3:
1. Hysteresis
This is the section that reviewer#2 holds strong objection, while reviewer#1 also wants the authors to clarify. I agree with the authors that these experiments are potentially interesting and provide valuable information for the research community, but some concerns (see below minor comments) still need to be addressed.

2. pCO2 reconstruction
As reviewer#2 points out that a drawback of the study is extending the simulation to the late Eocene (38 Ma) but completely ignoring the role of tectonics on CO2 in the model. I see the authors make this limitation clearer and show more quantified results of their simulated pCO2 compared with available proxies in the revised manuscript (in new Sec 3.3). In my view, more sophisticated model and setup are necessary to fully address such a caveat in the future studies.

Sec 4:
The ice-albedo feedback and the surface-height-temperature feedback are compared, and they have potentially interesting implications for the interpretation of ENSO proxy data. The overall argument under three CO2 scenarios (low CO2 -> strong NH ice-sheet and temperature change, albedo feedback more important; intermediate CO2 -> little melt NH ice-sheet and no deglaciation of Antarctica, minor temperature change; high CO2 -> strong SH ice-sheet and temperature change, elevation feedback more important) is fairly convincing.

We thank the reviewer for considering our manuscript, and we are pleased he/she agrees on the merit of our study. Below, we will reply to the minor concerns raised, and explain how we have revised the manuscript. We believe this has improved the quality of the paper, particularly of Section 3.1, and hope it is to the satisfaction of the reviewer.

Minor comments:

As suggested by reviewer#1, it is curious that a hysteresis behavior occurs when excluding overturning. For completeness the authors should at least list previous studies that has a similar feature but due to overturning (e.g. Zhang et al., 2014). Furthermore, after the discussion that roles out the possibility of coupled ice-sheet, coupled ocean overturning and sea ice and snow cover, the authors argue at the end of Sec 3.1 that 'this means that the hysteresis is inherent to the core of the climate model: the parameterisation of vertical and horizontal energy transfer in the ocean and atmosphere', but the conjecture is not really convincing. More evidences should be provided.

This concern is shared by reviewer #1. We now refer to the study mentioned (Zhang et al., 2014). Furthermore, we provide more evidence for our claim that the energy balance calculations are the cause of the hysteresis, by showing the energy imbalance at the top of the atmosphere. See also our reply to the comment of reviewer #1.

*Page 6, line 12-13:*
Keeping the ocean overturning strength, a known source of hysteresis in other models (e.g. Zhang et al., 2014), fixed at PD also leads to a small reduction; the difference becomes 0.73 K (Fig. 3c).

Also in the conclusion section it would be better for the authors not to simply list the conclusion as in the abstract, but to strengthen the connection between them (e.g. what's the implication of hysteresis and pCO2 caveat in Sec 3 that potentially influence the conclusion in Sec 4), and to more convincingly demonstrate that those materials in Sec 3 are crucial to comprehensively understand the target of the study.

We have expanded the conclusion section with some text, explaining why we retune the model because of the hysteresis. Furthermore, we briefly describe the implications of the retuning for the findings in Section 4, the influence of land ice on the ESS and polar amplification. This should help the reader to better understand the importance of Section 3.

*Page 12, line 26 – page 13, line 3:*
The hysteresis is most likely a consequence of the temperature dependency of the energy balance calculations. This is suggested by the fact that the model converges to a different steady-state energy imbalance at the top of the atmosphere, and therefore a different climate state, depending on the history of the run. It is still unknown whether this behaviour is also exhibited by other models. We therefore suggest that in the future, climate models should be tested for this behaviour by confronting them with high $CO_2$ values before simulating cooler climates. In our case, it leads to unrealistically low simulated $CO_2$ values over roughly the past 13 Myr in our long 38-Myr simulation when $CO_2$ levels are below 450 ppm, to compensate for the higher temperatures. We have therefore retuned the model, by changing the uncertain cloud optical thickness parameter. This leads to lower temperatures at the same $CO_2$ levels, such that higher $CO_2$ concentrations are sufficient to obtain the same temperatures. After retuning, the simulated $CO_2$ over the past 800,000 years is in reasonable agreement with the ice-core record again. The relations between $CO_2$ and temperature, as well as between high- and low-latitude temperatures, are now different. However, the influence of ice sheet-climate interactions on ESS and polar amplification remains qualitatively the same.

Editorial comments:

A few words and phrases (especially in the revised part) need to proof read as the English is not correct, e.g. P8L15, P8L26.

These lines have been revised. We will also opt for a proof read by the copy editor, which is hopefully still a service this journal provides.

Refs
Zhang, X., Lohmann, G., Knorr, G., & Purcell, C. (2014). Abrupt glacial climate shifts controlled by ice sheet changes. Nature, 512(7514), 290-294.

**Reviewer #3**
 Comments
This paper carried out transient simulations over the past 38 millions of years, an ice-sheet model coupled with an energy balance model. This work aims at studying how ice-sheets impact on climate changes in the past 38 million years.
Although the model is simple and idealized, this work is one of very few integrations over such a long time. The study is a good reference for future studies with mode sophisticate models. I noticed that it has taken a quite long period in paper reviewing and revising. I found that the authors have made sufficient changes in the revised version. Thus, I would recommend publication of the paper.

We thank the reviewer for considering our manuscript.

I read through the manuscript and did not find major problems.
There is one minor thing: page 11, line 7, "haven " –> "have".

The typo has been  corrected.

**Reviewer #4**
"The influence of ice sheets on the climate during the past 38 million years"
Author(s): Lennert B. Stap, Roderik S. W. van de Wal, Bas de Boer, Richard Bintanja, and Lucas J. Lourens

I was indeed very busy these last two months and, because I raised major criticism in my first review and also because the authors provided an improved manuscript, it took me some time to reconsider this manuscript. Therefore, I apologize for this long delay.

General comments:
A very good point when reviewing a CP paper is that the different reviews of the paper are published and therefore are available to the readers. I did not change my mind concerning this paper. Indeed, as I said in my first review, this is an interesting, but limited step towards capturing ice sheet impact on climate evolution. Interesting, because it is using a simplified tool that enables the author to explore ice sheets and climate evolution since 38 Ma but limited, because the comparison with real evolution of climate due to the use of EBM for climate and 1D ice sheet model for ice sheets that only allow for a very theoretical approach. Moreover, even if the structure of the paper has been improved, I think that a first step using realistic $CO_2$ reconstructions would have been a very interesting exercise, rather than reconstructing $CO_2$, which is anyway a very difficult exercise. The authors now answer to my major comments concerning the weakness of their introduction in terms of comparison with previous studies. Also, as requested by the first reviewer, they added more information, included equations on their model and clarified their experimental design and comparison to $CO_2$ reconstruction. They did not convince me on some aspects (see detailed review below) but I nevertheless consider that, with all the limitations they already included and, accounting for suggestions and remarks below, this paper may be published in Climate of the Past.

We thank the reviewer for another careful consideration of our manuscript. We are pleased that the reviewer now finds it worthy of publication after minor revisions. Below, we describe how we have dealt with his/her remaining concerns. We believe this has improved the quality of the manuscript.

Detailed comments:
title: I think the title should refer to the fact it is a modeling study. Therefore, I would suggest "Influence of ice sheets on the climate during the past 38 million years inferred from a simple coupled ice sheet/climate model". This point is crucial for me, because, due to many major simplifications in this approach, the results are very theoretical, compared to the real climate evolution.

We have changed the title to "The influence of ice sheets on temperature during the past 38 million years inferred from a one-dimensional ice sheet-climate model".

Methodology:
This part has been clearly improved due to both reviewer comments. Nevertheless, I think that, because the authors have a simple tool, that enables them to perform many simulations accounting for different CO2 scenarios, it would have been very interesting to prescribe reconstructed CO2 evolutions from different proxies to explore ice-sheet and climate evolution. Especially, to analyze the Antarctica inception, as simulated in the different scenarios, (e.g. when it occurred, and for which CO2 level, without any tuning). Nevertheless, I agree that they use a consistent approach, but tuning parameters, without understanding where are the bias and caveat of this method.
I am not sure I fully understand for instance why they have to tune the CO2 value for inception to the one from Pollard and Deconto and what are the consequences of this tuning.

The tuning targets for the ablation threshold parameters of the different ice sheets have been described in an earlier publication (Stap et al., 2014). For the East Antarctic ice sheet, the model was tuned to produce a realistic present-day ice sheet. Since this was the case in a large range of values for the ablation parameter, the extra constraint of glacial inception at roughly 700 ppm was used. The effect of altering this parameter for the EAIS on the inception of this ice sheet has also been discussed before (Stap et al., 2016), although maybe not in the amount of detail the reviewer would have liked. It was shown that changing the value of $C_{abl}$ from -30 to -5 results in glacial inception at roughly 450 ppm instead of 700 ppm. We now refer to these publications in the model section. We feel, however, that the analysis the reviewer suggests would be partly repetitious, and in any case beyond the scope of this paper. We refrain from using prescribed $CO_2$-scenarios, because we feel the difference between our inversely simulated $CO_2$ and proxy data, most importantly the lack of a simulated difference between MMCO and EOT $CO_2$ levels, already reflects the caveats in our model approach very accurately. The inverse approach we take here, is in our opinion a natural continuation of a long line of papers, the most important ones being listed in the introduction section.

*Page 4, line 15-18:*
The tuning targets and consequent values for $C_{abl}$ have been described in Stap et al. (2014). The influence of this parameter for inception of the East Antarctic ice sheet has been discussed in Stap et al. (2016b), who showed that changing the value from -30 to -5 results in glacial inception at roughly 450 ppm instead of 700 ppm.

Discussion:
In the discussion, the authors argue on the impact of ice sheets variability on climate for high and low CO2. It is a very consistent, but theoretical analysis, when compared to real climate evolution for many reasons, some are related to the fact that it is only the radiative forcing and not the dynamics that is considered here.

On the other hand, rapid dynamics change of the ice sheets may have important consequences on tropical atmospheric circulation (see for instance Mulitza 2008 for Heinrich events and African monsoon) that may change drastically our view on the variability of the ice sheet and its impact during cold and warm phases.

Our focus is on long-term interactions, which is mentioned several times in the manuscript. We have added a sentence to the conclusion section, admitting to our models' inability to capture sub-millennial climate dynamics. We recognize, in agreement with reviewer #2, that studies with more detailed models remain important to simulate smaller spatial and temporal scales.

*Page 12, line 14-16:*
The physics in our model are not detailed enough to simulate sub-millennial climate dynamics, which could have important consequences on for instance tropical atmospheric circulation (Mulitza et al., 2008).

Comments on figures:
Fig 4: Please specify that the red curve is the new reference. The x axis title should be (Myears).
Fig 4A: the new ref systematically underestimates the values usually accounted for this period, after mis-M2 (~3.3 Ma) and till 3.0 Ma, for this warm period, PLIOMIP simulations take a value of 405 ppm (Haywood 2012), significantly larger than the new reference results (300 ppm).

The figure and caption have been corrected. A reference to the PlioMIP experiments is included in the text.

*Page 8, line 15-18:*

The variability in our simulation is more in line with the boron isotope proxy (Martínez-Botí et al., 2015), although our simulated $CO_2$ values are lower between 3.2 and 2.8 Myr ago, closer to the alkenone estimates Badger et al. (2013). Modelling studies with General Circulation Models (GCMs) generally use higher values for the Pliocene, for example 400 ppm in the forthcoming PlioMIP2 experiments (Haywood et al., 2016).

Fig 4b: For MMCO, the CO2 are overestimated, with respect to the data.
Fig 4c: In this case, the new reference results are lower than the data. This figure illustrates the interest of providing simulations with consistent proxy reconstruction (alkenone, boron, stomates) rather than computed CO2, which is either overestimating or underestimating these reconstructions and moreover did not capture correctly all the processes for this geological time span.

Please refer to our reply to an earlier comment of this reviewer.

Fig. 5: It would be very interesting to make several plots rather than only one with the three computed parameters to compare also the sea level evolution and the temperatures to data. For instance, 5a could be delta O18 and CO2 and 5b could be sea level evolution computed and compared to data and 5c temperature computed and compared to data.

We prefer to show only our results in this figure, because it is clearer. We already show comparisons to other data and model results in figures 2, 5, 7 and supplementary figure 3. Our modelled sea level and temperature are largely in agreement with De Boer et al. (2010) and De Boer et al. (2012), who have done further comparisons. For instance, they compared their ice volume simulation to sea level data from Red Sea sediments (Rohling et al., 2009), and their temperature simulation to a Mg/Ca proxy data deep-sea temperature record (Sosdian and Rosenthal, 2009b). We now refer the reader to this earlier work.

*Page 9, line 5-8:*

[revised manuscript text omitted]